# Bik reduces hyperplastic cells by increasing Bak and activating DAPk1 to juxtapose ER and mitochondria

Yohannes A. Mebratu[1], Ivan Leyva-Baca[1], Marc G. Wathelet[2], Neal Lacey[1], Hitendra S. Chand [1],
Augustine M.K. Choi[3] & Yohannes Tesfaigzi[1]

Bik reduces hyperplastic epithelial cells by releasing calcium from endoplasmic reticulum stores and causing apoptosis, but the detailed mechanisms are not known. Here we report that Bik dissociates the Bak/Bcl-2 complex to enrich for ER-associated Bak and interacts with the kinase domain of DAPk1 to form Bik–DAPk1–ERK1/2–Bak complex. Bik also disrupts the Bcl2–IP$_3$R interaction to cause ER Ca$^{2+}$ release. The ER-associated Bak interacts with the kinase and calmodulin domains of DAPk1 to increase the contact sites of ER and mitochondria, and facilitate ER Ca$^{2+}$ uptake by mitochondria. Although the Bik BH3 helix was sufficient to enrich for ER-Bak and elicit ER Ca$^{2+}$ release, Bik-induced mitochondrial Ca$^{2+}$ uptake is blocked with reduced Bak levels. Further, the Bik-derived peptide reduces allergen- and cigarette smoke-induced mucous cell hyperplasia in mice and in differentiated primary human airway epithelial cultures. Therefore, Bik peptides may have therapeutic potential in airway diseases associated with chronic mucous hypersecretion.

[1] COPD Program, Lovelace Respiratory Research Institute, Albuquerque, NM 87108, USA. [2] Infectious Diseases Program, Lovelace Respiratory Research Institute, Albuquerque, NM 87108, USA. [3] Department of Medicine, Weill Cornell Medical Center, New York, NY 14850, USA. Correspondence and requests for materials should be addressed to Y.T. (email: ytesfaig@lrri.org)

FN-γ by activating STAT1[1] increases the susceptibility of cancer cells to apoptosis[2] and plays an important role in the removal of hyperplastic epithelial cells to control chronic mucous secretions in bronchitic asthma or chronic bronchitis[3–5]. IFN-γ sensitizes airway epithelial cells (AECs) to cell death[6] by increasing expression of the Bcl-2 interacting killer (Bik) and blocking nuclear translocation of ERK1/2[5]. Bik, being anchored in the endoplasmic reticulum (ER) initiates a Bak-dependent release of ER $Ca^{2+}$ stores[7], resulting in DRP1-regulated mitochondrial fission and release of cytochrome $c$ to initiate apoptosis[8]. However, the physiological stimuli that enrich Bak at the ER and which other proteins facilitate $Ca^{2+}$ transfer from ER to mitochondria are not known.

The ER is the main storage site for $Ca^{2+}$ within the cell. Inositol phosphate 3 (IP3)-dependent release of $Ca^{2+}$ from the ER into the cytoplasm produces $Ca^{2+}$ signals with diverse cellular functions such as cell proliferation and survival[9]. While $Ca^{2+}$ oscillations support cell survival in part by positively regulating mitochondrial metabolism, prolonged high-amplitude $Ca^{2+}$ release into mitochondria via the inositol 1,4,5-trisphosphate receptors (IP$_3$Rs)[10] causes $Ca^{2+}$ overload and apoptosis[11, 12]. The ER and mitochondria provide compartmentalized microenvironments, but these compartments communicate and exchange metabolites that ultimately determine the function of the cell. Proteins localized to the ER or mitochondria can determine sites of close contact also referred to as mitochondria-associated ER membrane. For example, mitofusin 2 (Mfn2) binds to ER derivatives of Mfn1 at specialized ER-mitochondrion contact sites[13] and the mitochondrial outer membrane (MOM) fission protein, Fis1, makes contact with ER-localized BAP-31[14], suggesting that there is a bi-directional communication between the two organelles. The macromolecular complexes that facilitate ER/mitochondria contact to determine between adaptive responses vs. proapoptotic signals have yet to be identified.

Other Bcl-2-related proteins also play a major role in regulating ER $Ca^{2+}$ levels[15] because enforced expression of Bak and Bax provokes ER $Ca^{2+}$ release[16, 17], and Bak/Bax can localize to the ER[17, 18] to regulate ER calcium levels in the reticular lumen[19]. In contrast, Bcl-2 overexpression prevents the reduction of ER $Ca^{2+}$ concentrations by its BH4 domain binding the regulatory and coupling domain of the IP$_3$R and inhibiting IP$_3$-dependent channel opening[20–23].

In the present study, we identified the proteins that Bik assembles to initiate ER $Ca^{2+}$ release and to facilitate efficient transfer to mitochondria. Bik increased Bak levels to enrich ER-associated Bak and facilitate the formation of the Bik–DAPk1–ERK1/2–Bak (BDEB) complex. We show that Bak is required for anchoring DAPk1 to the ER and increase the contact sites between ER and mitochondria to elicit transfer ER $Ca^{2+}$ to mitochondria. Bik also disrupts Bcl-2 and IP$_3$R interaction and causes ER-$Ca^{2+}$ release. A double hydrocarbon-stapled (DHS) peptide modeled after the Bik BH3 helix and does not include the ER-anchoring domain caused efficient Bak activation and cell death. Bik BH3 peptide restored cell death and reduced allergen- or cigarette smoke (CS)-induced epithelial and mucous cell hyperplasia in primary human AECs in culture and in vivo similar to the whole Bik protein when transgenically expressed in an inducible manner in airway epithelia of adult mice. Thus, Bik BH3 helix may be useful as a therapeutic agent to reduce mucous hypersecretion.

## Results

### Bak plays a central role in IFN-γ- and Bik-induced cell death.
IFN-γ causes resolution of hyperplastic epithelial cells in asthma by inducing apoptosis in AECs[3]. IFN-γ does not affect Bax expression[24], and *bax*-deficient cells are sensitive to IFN-γ-induced cell death[5]. However, IFN-γ increased the protein levels of the other major proapoptotic protein Bak[18, 25] in HAECs (Fig. 1a), primary HAECs (Supplementary Fig. 1a), and murine AECs (MAECs) (Supplementary Fig. 1b). The Leu61 within the BH3 domain of Bik was crucial in increasing Bak expression in HAECs (Fig. 1b). Similarly, Ad-Bik increases Bak expression (Supplementary Fig. 1c) and causes cell death (Supplementary Fig. 1d) in primary mouse colonic epithelial cells that also show goblet cell hyperplasia.

Because IFN-γ induces Bik expression in AECs in a STAT1-dependent manner[5], we investigated whether STAT1 or Bik mediate IFN-γ-induced Bak expression. Loss of STAT1 (Supplementary Fig. 1e) or Bik (Fig. 1c), but not Noxa (Supplementary Fig. 1f) abrogated IFN-γ-induced Bak protein expression. Presence or absence of Bik did not affect IFN-γ-induced Bak mRNA levels (Supplementary Fig. 1g); therefore, we tested whether expression of Bik stabilizes Bak by abrogating its interaction with Bcl-2. Suppression of Bcl-2 using shBcl-2 or treatment with IFN-γ increased Bak protein levels (Fig. 1d). Also, inhibition of the proteasomal degradation using MG-132 increased Bak protein levels in shCtrl but further enhanced Bak levels in shBcl-2 cells (Fig. 1d). Expression of Bik$^{WT}$ but not Bik$^{L61G}$ dissociated Bcl-2 from BAK (Fig. 1e). These findings suggest that Bak when present as Bcl-2/Bak complex is degraded more rapidly after disruption of Bak/Bcl-2 complex by Bik. We found that both IFN-γ at 24 (Supplementary Fig. 1h) and at 72 h (Supplementary Fig. 1i), or expression of Bik but not mutant Bik (Fig. 1f) activated Bak in HAECs as probed using conformation-specific anti-active Bak antibody by flow cytometry.

### Bik enriches Bak at the ER to elicit ER $Ca^{2+}$ release.
Bik is anchored on the ER membrane, promotes ER calcium release to cause mitochondrial apoptosis[7], and while Bak is constitutively anchored on the MOM[26], it can also be localized to the ER[17, 18]. As a phospho-protein[27], Bak gets activated by a conformational change in the N-terminus following dephosphorylation at tyrosine 108 (Y180)[28]. Bik increases Bak protein levels within the ER fraction (Supplementary Fig. 1j), which was visualized by multimers in ER fractions of both Bak$^{WT}$- and Bak$^{Y108A}$-expressing cells, but more pronounced in Bak$^{Y108A}$-expressing cells (Fig. 1g).

Confocal microscopy showed that in HAECs treated with IFN-γ (Supplementary Fig. 1k) or Ad-Bik (Supplementary Fig. 1l), the number of cells with activated Bak co-localized to the ER increased by 3–5-fold. Suppression of Bik using shRNA (Supplementary Fig. 1m) reduced the number of IFN-γ-treated cells with Bak co-localized to the ER (Fig. 1h). Together, these findings suggest that Bik mediates IFN-γ-induced Bak localization to the ER. Further confirmation for the translocation of Bak to the ER was derived from IFN-γ-treated HCT116$^{bax-/-bak-/-}$ cells expressing fluorescently labeled F3YpetR-F3hBak. HCT116$^{bax-/-bak-/-}$ cells were protected from IFN-γ (Supplementary Fig. 1n) or Ad-Bik (Supplementary Fig. 1o)-induced cell death. In addition, analysis of co-localization using the Manders' coefficient showed that IFN-γ caused significant translocation of F3YpetR-F3hBak to the ER (Fig. 1i).

We further investigated the hypothesis that Bak accumulation on the ER may affect ER calcium release ([$Ca^{2+}$]$_i$). IFN-γ treatment or Ad-Bik expression caused release of ER $Ca^{2+}$ just 4–6 h after treatment (Fig. 1j). To determine whether Bik disrupts Bcl-2–IP$_3$R interaction that is known to regulate the flux of ER $Ca^{2+}$[29], we infected HAECs with Ad-Bik or Ad-Bik$^{L61G}$ and co-immunoprecipitated the protein lysates with anti-IP$_3$R or anti-Bcl-2 antibodies. Expression of Bik$^{WT}$ but not Bik$^{L61G}$ disrupted the Bcl-2–IP$_3$R interaction (Fig. 1k). Because Bak-deficient cells were resistant to IFN-γ- or Ad-Bik-induced cell death, we tested

whether deficiency of Bak also impairs Ad-Bik-induced ER $[Ca^{2+}]_I$ efflux and $[Ca^{2+}]_m$ accumulation. The number of cells with ER $[Ca^{2+}]_I$ efflux (Fig. 1l) and mitochondrial calcium $[Ca^{2+}]_m$ (Supplementary Fig. 1p) accumulation was 6–7-fold higher in $bak^{+/+}$ compared with $bak^{-/-}$ MAECs when Bik was expressed, suggesting that ER-Bak is required for ER $[Ca^{2+}]_I$ efflux.

**Bak mediates resolution of hyperplastic epithelial cells.** MAECs from $bak^{-/-}$ mice were significantly more resistant to 50 ng/ml murine IFN-γ over 24, 48, and 72 h of treatment (Fig. 2a) or Bik expression (Fig. 2b). Interestingly, even $bak^{+/-}$ MAECs were protected from Ad-Bik-induced cell death (Fig. 2b), suggesting that the dosage of Bak protein is important for IFN-γ and Bik-

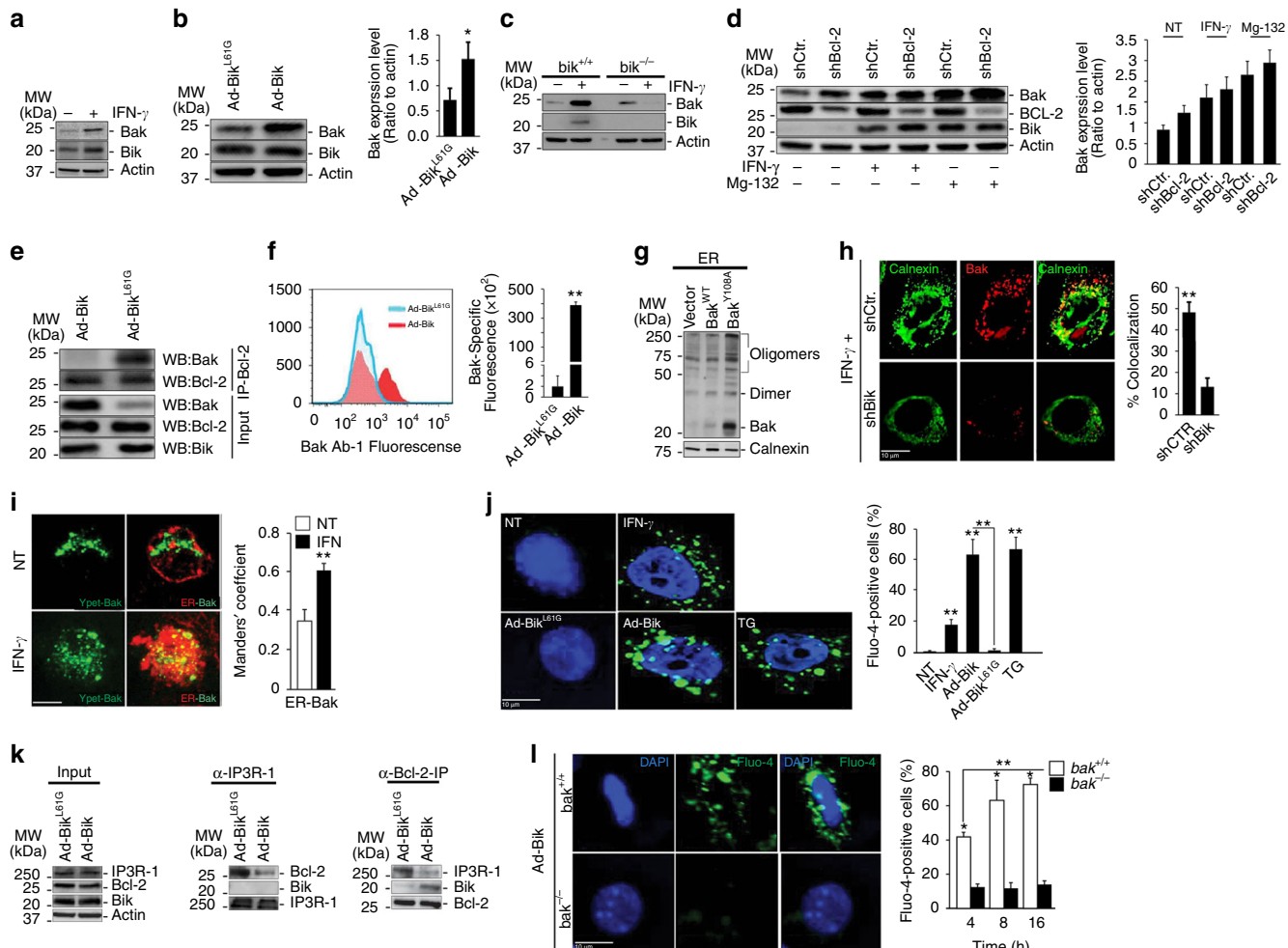

**Fig. 1** Bik activates and translocates Bak to the ER. HAECs were treated with 50 ng/ml human recombinant IFN-γ or medium alone as controls (**a**) or infected with 100 MOI Ad-Bik or Ad-Bik$^{L61G}$ (**b**), and 24 h later protein lysates were analyzed for changes in the expression of Bak by western blotting. The fold change in Bik level was analyzed by densitometry. **c** MAECs from $bik^{+/+}$ or $bik^{-/-}$ mice were treated with IFN -γ for 48 h and the levels of Bik and Bak protein levels analyzed by western blotting. **d** HAECs were transiently infected with retroviral vectors for shCtr or shBcl-2 and treated with 50 ng/ml IFN-γ or medium alone as control for 48 h or treated with 10 μM MG132 for 24 h and protein lysates were analyzed for the expression level of Bak, Bcl-2, Bik and actin proteins by western blotting. **e** HAECs were infected with 100 MOI Ad-Bik or Ad-Bik$^{L61G}$, and immunoprecipitates of protein lysates with anti-Bcl-2 antibody were probed for Bak and Bcl-2 levels by western blotting. **f** HAECs were infected with Ad-Bik or Ad-Bik$^{L61G}$ and 24 h later analyzed for increases in active Bak using the Ab-1 antibody that is specific for activated Bak using FACS. **g** HCT116 cells retrovirally transduced with empty vector, Bak$^{WT}$ or Bak$^{Y108A}$ were infected with 100 MOI Ad-Bik. The ER fractions were analyzed with anti-Bak antibodies. Calnexin was used as a marker for equal loading of ER proteins. **h** Representative photomicrographs of HAECs stably expressing shCtr or shBik treated with 50 ng/ml IFN-γ for 48 h fixed with paraformaldehyde and immunostained for activated Bak and calnexin. Percent of cells with Bak localized to the ER was quantified after counting at least 200 cells for each experiment. Scale bar, 5 μm. **i** Representative micrographs of HCT116$^{bax-/-bak-/-}$ cells transfected with F3YpetR-F3hBak (green), followed by treatment with 50 ng/ml IFN-γ or media alone. Cells were stained with calnexin for ER (red) and confocal images of 3D reconstructions of ER were acquired. Areas of Bak-ER co-localization were quantified by Manders' coefficient, means and SEM (n = 3 experiments, 40 cells per experiment). Scale bars, 5 μm. **j** HAECs were treated in the presence of 20 μM pan-caspase inhibitor Q-VD-OPh with IFN-γ or nothing, thapsigargin (positive control), Ad-Bik or Ad-Bik$^{L61G}$ for 4 h, and stained with Ca$^{2+}$-flux indicator, Fluor-4 (green) and counterstained with DAPI (blue) for nuclei. **k** HAECs were infected with 100 MOI Ad-Bik or Ad-Bik$^{L61G}$ and 24 h later protein lysates immunoprecipitated using anti- IP$_3$R-1 or anti-Bcl-2 antibodies. Input and immunoprecipitates were probed for Bcl-2, Bik, and IP$_3$R-1 levels by western blotting. **l** Deficiency of bak impairs Ad-Bik-induced ER Ca$^{2+}$ efflux. MAECs from $bak^{+/+}$ or $bak^{-/-}$ mice were infected with Ad-Bik for 4, 8 and 16 h, and stained with Ca$^{2+}$-efflux indicator, Fluo-4. (n = 3 experiments, with >200 cells analyzed per condition). Differences between two groups were assessed for significance by Student's t test. ANOVA was used to perform pair-wise comparison of the data from more than two groups followed by Fisher least significant difference test. Graphs show mean ± SEM; * = P < 0.05; ** = P < 0.01

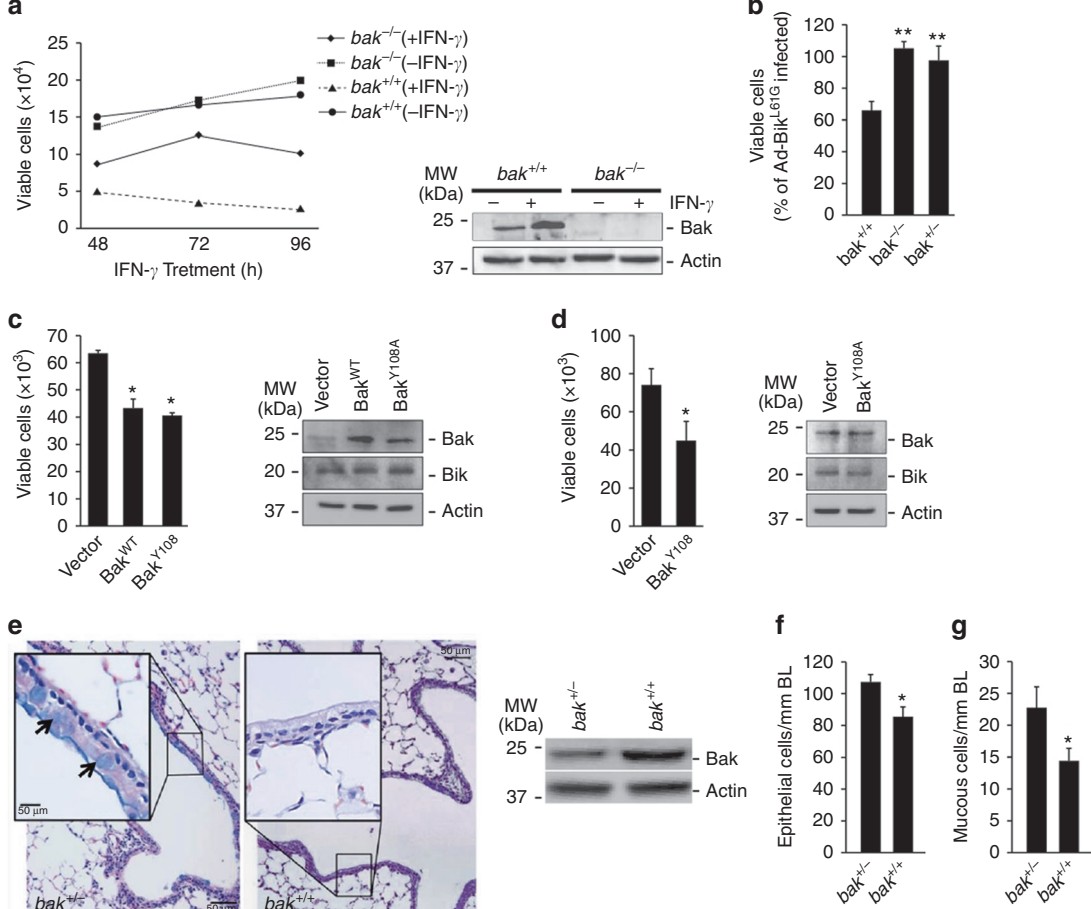

**Fig. 2** Bak mediates IFN-γ- or Bik-induced cell death and resolution of ECH and MCM during prolonged exposure to allergen. **a** MAECs isolated from $bak^{-/-}$ and $bak^{+/+}$ mice treated with 50 ng/ml murine recombinant IFN-γ or left untreated as controls and viable cells quantified 24, 48, and 72 h later with Trypan blue. Protein lysates were analyzed for the expression level of Bak by western blot. **b** MAECs from $bak^{+/+}$, $bak^{-/-}$, and $bak^{+/-}$ mice were infected with 100 MOI Ad-Bik or Ad-Bik$^{L61G}$, and cell viability was quantified 24 h later. **c** MAECs from $bak^{-/-}$ mice were infected with empty vector or retroviral expression vector for Bak$^{WT}$ or mutant constitutively activation competent form of Bak (Bak$^{Y108A}$). Cells were collected and quantified by trypan blue exclusion assay and protein lysates were analyzed by western blotting. **d** MAECs from $bak^{+/-}$ mice were infected with empty vector and retroviral expression vectors for mutant constitutively activation competent form (Bak$^{Y108A}$) of Bak. And 48 h later, cells were infected with 100 MOI Ad-Bik for 24 h. Cells were collected and quantified by trypan blue exclusion assay and protein lysates were analyzed by western blotting. **e** Representative micrographs of $bak^{+/+}$ and $bak^{+/-}$ mice that were immunized with ovalbumin/alum on d 1 and 7, and were exposed to ovalbumin aerosols for 15 d. After killing, the lungs of each mouse were fixed and tissue sections were stained with Alcian blue/hematoxylin and eosin (AB/H&E) for quantification. Protein lysates of airway epithelial cells from $bak^{+/-}$ and $bak^{+/+}$ mice were analyzed for Bak protein levels by western blotting. The numbers of epithelial **f** and mucous **g** cells per millimeter of basal lamina in $bak^{+/+}$ mice but not in $bak^{+/-}$ mice at 15 d of exposure. Differences between two groups were assessed for significance by Student's $t$ test. ANOVA was used to perform pair-wise comparison of the data from more than two groups followed by Fisher least significant difference test. Error bars indicate ± SEM; ($n = 8$ mice per group). * = $P < 0.05$; ** = $P < 0.01$

induced cell death. Further, expression of Bak$^{WT}$ or Bak$^{Y108A}$ using a retroviral vector restored Ad-Bik-induced cell death of $bak^{-/-}$ (Fig. 2c) and $bak^{+/-}$ (Fig. 2d) MAECs, demonstrating that Bak is crucial for IFN-γ- and Bik-induced cell death pathway. IFN-γ (Supplementary Fig. 2a) and Ad-Bik (Supplementary Fig. 2b) caused significant increases in Annexin V positivity compared to the respective controls, confirming that this pathway induces apoptosis.

IFN-γ through Bik mediates the resolution of hyperplastic cells, when mice are exposed to allergen for a prolonged period[5]. Therefore, we investigated the physiological relevance of Bak in vivo by testing whether this resolution process would be abrogated in $bak^{+/-}$ mice. Allergen-induced epithelial cell hyperplasia (ECH) remained significantly higher in $bak^{+/-}$ compared with $bak^{+/+}$ mice after 15 days of allergen exposure (Fig. 2e, f). Furthermore, the number of mucous cells per millimeter of basal lamina (BL) was sustained in the lungs of

$bak^{+/-}$ mice, while significantly reduced in $bak^{+/+}$ mice (Fig. 2e, g), demonstrating that Bak is an obligatory protein for Bik-induced cell death in vivo.

**DAPk1 assembles Bak and ERK1/2 to mediate Bik-induced death.** Bik interacts with and blocks nuclear localization of ERK1/2 to promote cell death[5, 30] and ERK1/2 promotes the apoptotic activity of DAPk1 by interacting with the death domain[31]. Therefore, we tested the hypothesis that Bik forms a complex with DAPk1 and ERK1/2. We expressed HA-tagged Bik or Bik$^{L61G}$ in HAECs and detected both Bik and DAPk1 in the immunoprecipitates with anti-p-ERK1/2 antibodies in cell expressing Bik but not Bik$^{L61G}$ (Fig. 3a). The site of interaction of Bik with DAPk1 was further examined by immunoprecipitation with anti-Flag antibodies of protein lysates from 293T cells that were transfected with Flag-DAPk1 and infected with adenoviral vectors expressing

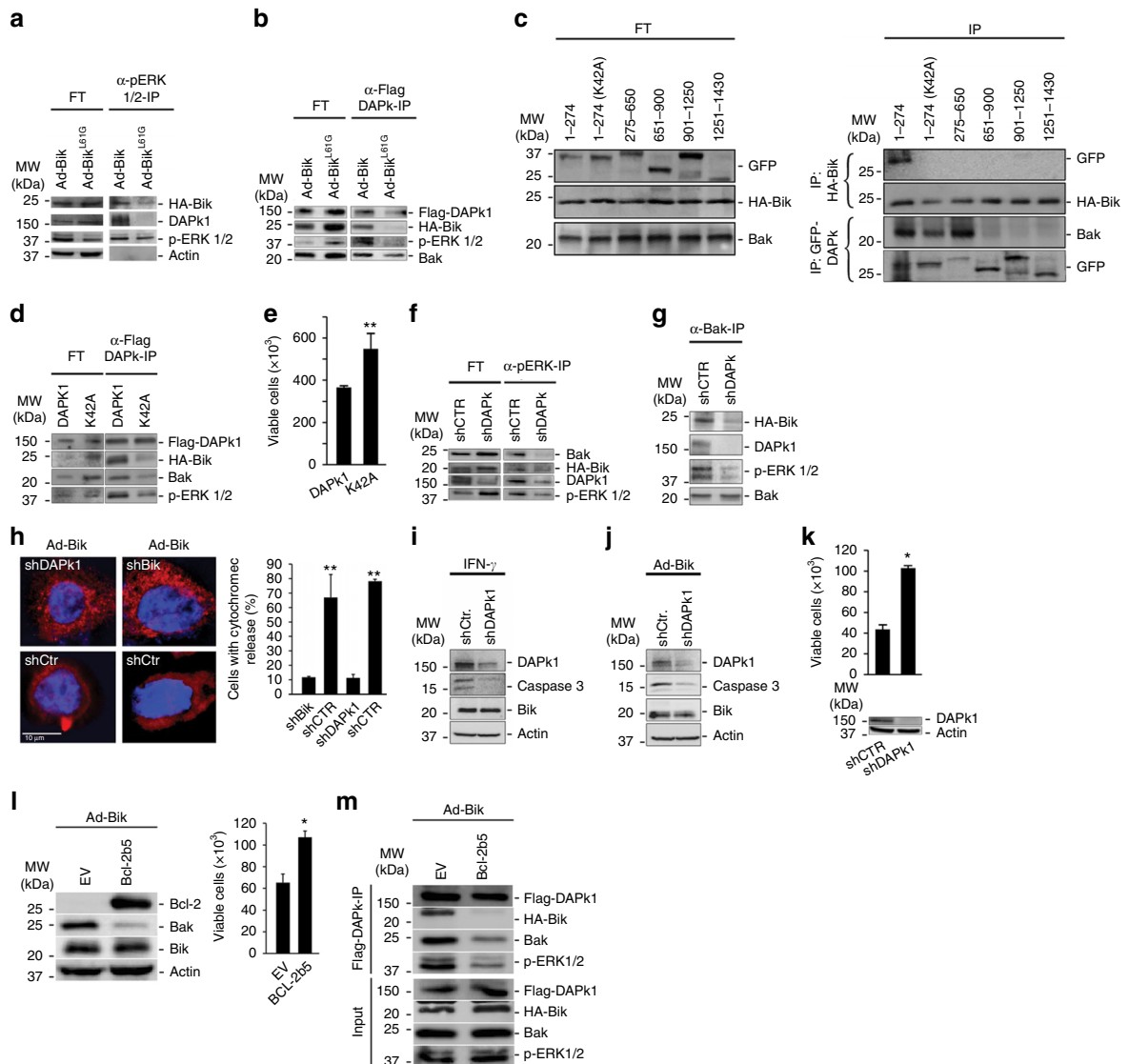

**Fig. 3** DAPk1 facilitates Bik-induced cell death by forming a complex with Bik, ERK1/2 and Bak. **a** Protein lysates from HAECs 24 h after infection with HA-Ad-Bik or HA-Ad-Bik[L61G] were immunoprecipitated with anti-pERK1/2 antibodies. The flow through (FT) and the immunoprecipitates (IPs) were probed with antibodies to HA, DAPK1, p-ERK1/2, and ERK1/2 by western blotting. **b** 293T cells were transfected with Flag-DAPk1 and 48 h later infected with 100 MOI HA-Ad-Bik or HA-Ad-Bik[L61G]. After 24 h, protein lysates were immunoprecipitated with α-Flag antibodies and the FT and IP were probed for activated Bak, HA, and ERK1/2 by western blotting. **c** 293T cells transfected with various GFP-tagged DAPk1 deletions constructs, infected with HA-Ad-Bik, and subjected to immunoprecipitation using anti-HA or anti-GFP antibodies. The flow through (FT) and the IPs were probed for GFP, HA, activated Bak.
**d** Mutation of Lys42 in the kinase domain of DAPk1 diminishes Bik–Bak–DAPk1–ERK1/2 interaction. 293T cells were transfected with Flag-DAPk1 or Flag-DAPK1[K42A] and 48 h later infected with 100 MOI HA-Ad-Bik. Protein lysates were immunoprecipitated using α-Flag antibodies and the FT and IP were probed for Flag, HA, Bak, and ERK1/2 by western blotting. **e** Cell viability of 293T cells transfected with Flag-DAPk1 or Flag-DAPK1[K42A] was analyzed by trypan blue assay. **f, g** Knockdown of DAPk1 suppresses the interaction between Bak, Bik, and ERK1/2. Protein lysates from HAECs stably expressing shCtr or shDAPk1 were infected with HA-Ad-Bik and immunoprecipitated using anti-pERK1/2 **f** or anti-active Bak (Ab1) **g** antibodies. The FT and IPs were probed for activated Bak, HA, DAPk1, and p-ERK1/2 by western blotting. **h** HAECs stably expressing shCtr or shDAPk1 were infected with 100 MOI of Ad-Bik in the presence of 20 μM pan-caspase inhibitor Q-VD-Oph for 18 h, fixed and stained for cytochrome c, and the percentage of cells with cytochrome c release was analyzed by fluorescent microscopy. **i, j** Cell lysates from HAECs stably expressing shCtr or shDAPk1 and treated with 50 ng/ml IFN-γ **i** or 100 MOI Ad-Bik **j** were analyzed for cleaved caspase 3 by immunoblotting. **k** HAECs stably expressing shCtr or shDAPk1 were analyzed for knockdown of DAPk1 by western blotting. Cell viability was determined by trypan blue exclusion assay 24 h after infection with100 MOI Ad-Bik. **l** 293T cells were transfected with plasmids expressing empty vector (EV) or ER-targeted Bcl-2 (Bcl-2b5). Forty-eight hours later, cells were infected with 100 MOI Ad-Bik and protein lysates were analyzed for the expression of Bcl-2, Bak, and Bik. Cell viability was determined by trypan blue exclusion. **m** 293T cells were transfected with plasmids expressing empty vector or Bcl-2b5 together with plasmids expressing Flag-DAPk1 and 48 h later infected with 100 MOI HA-Ad-Bik. Protein lysates were immunoprecipitated using α-Flag antibodies and the FT and IP were probed for Flag-DAPk1, HA-Bik, Bak and ERK1/2 by western blotting. Differences between two groups were assessed for significance by Student's *t* test. ANOVA was used to perform pair-wise comparison of the data from more than two groups followed by Fisher least significant difference test. Error bars indicate ± SEM, $n = 5$; * = $P < 0.05$, ** = $P < 0.01$

HA-Bik or HA-Bik$^{L61G}$. DAPk1 interacted with Bik, Bak and ERK1/2, and mutation of the Bik BH3 domain at L61G diminished these interactions (Fig. 3b), suggesting that the Bik BH3 domain interacted with DAPk1 and that Bik by interacting with DAPk1 initiates interaction with Bak and ERK1/2.

DAPk1 is a Ser/Thr kinase that is localized to actin microfilaments and contains several functional domains[32]. Therefore, we generated GFP-tagged expression constructs that encode for the following domains: amino acids (aa) 1–274 encoding for the kinase domain, aa 275–650 for the $Ca^{2+}$-calmodulin (CaM) domain, aa 651–900 for the ankyrin repeats, aa 901–1250 for the cytoskeletal domain, and aa 1251–1430 for the death domain motif. These GFP-tagged deletion constructs were transfected into 293T cells and HA-tagged Bik expressed using the adenoviral vector. Western blotting of the anti-HA immunoprecipitates from lysates of transfected cells revealed that Bik interacts with the kinase domain of DAPk1, and mutation of the kinase domain (K42A) impaired this interaction (Fig. 3c). Interestingly, Bak interacted with both kinase- and CaM-binding domains of DAPk1 and mutation of the kinase domain of DAPk1 diminished but did not fully disrupt the Bak/DAPk1 interaction (Fig. 3c). To further test the role of the kinase domain of DAPk1 in modulating Bik-induced cell death, we transfected 293T cells with Flag-DAPk1 or Flag-DAPk1$^{K42A}$ and 48 h later expressed HA-Bik using adenoviral vector. Mutation of lysine to alanine (K42A) in the kinase domain of DAPk1 impaired its interaction with Bik and Bak (Fig. 3d) and diminished Bik-induced cell death (Fig. 3e). Suppression of DAPk1 with shRNA impaired interaction of Bik, Bak, and ERK1/2 both after immunoprecipitation with anti-pERK1/2 (Fig. 3f) or anti-Bak antibodies (Fig. 3g).

Significant release of cytochrome c from the mitochondria was observed at 8 h after Ad-Bik infection, with the majority of cells displaying cytochrome c release at 18 h (Supplementary Fig. 3a). Cytochrome c release was also detected in the cytosolic fraction of lysed cells (Supplementary Fig. 3b). However, the Ad-Bik-induced cytochrome c release was significantly diminished when DAPk1 or Bik were suppressed (Fig. 3h). Further, caspase 3 was activated by IFN-γ (Supplementary Fig. 3c) or by Ad-Bik (Supplementary Fig. 3d), and this activation was diminished significantly in cells with suppressed DAPk1 (Fig. 3i, j).

Previous reports showed that loss of Bik did not protect hematopoietic cells from apoptosis[33], although we found that Bik was crucial for epithelial cells[5]. To determine the possible role of DAPk1 in these observed differences, we screened different cell types for expression levels of DAPk1 protein (Supplementary Fig. 3e). Cells with lower DAPk1 protein expression (Supplementary Fig. 3f) were less sensitive to Bik-induced cell death (Supplementary Fig. 3g), suggesting that DAPk1 levels are crucial for Bik-induced cell death. The Bik-induced cell death (Fig. 3k) and Annexin V positivity (Supplementary Fig. 3h) were reduced when DAPk1 expression was suppressed in HAECs using shDAPk1 construct. To confirm whether increased expression of ER-Bcl-2 abrogates Bik-induced formation of BDEB complex, we over-expressed ER-targeted Bcl-2 (Bcl-2b5) in 293T cells and analyzed whether Bik-induced cell death and the formation of BDEB complex is affected. We found that both Bik-induced Bak activation and cell death (Fig. 3l), and BDEB formation (Fig. 3m) were impaired in cells expressing Bcl-2b5. Together, these results suggest that DAPk1 after interacting with Bik assembles ERK1/2 and Bak at the ER to facilitate cytochrome c release and cell death.

**DAPk1 tethers ER to mitochondria to facilitate $Ca^{2+}$ release.** The efflux of ER calcium ($[Ca^{2+}]_i$) and elevation of mitochondrial $Ca^{2+}$ ($[Ca^{2+}]_m$) were analyzed using a fluorescent-based ER and mitochondrial calcium indicators, respectively. Bik expression

caused the release of ER $Ca^{2+}$ (Fig. 4a, Supplementary Fig. 4a), and increase of mitochondrial $[Ca^{2+}]_m$ (Fig. 4b), which were diminished significantly when DAPk1 levels were suppressed in shDAPk1 cells. Further, suppression of Bik using shRNA inhibited IFN-γ- (Supplementary Fig. 4b) or Ad-Bik (Supplementary Fig. 4c, d)-induced ER-$Ca^{2+}$ release and mitochondrial $Ca^{2+}$ accumulation (Supplementary Fig. 4e).

$Ca^{2+}$ transmission from the ER to mitochondria occurs at contact sites between the two organelles. However, little is known about how physiological stimuli control the distance between ER and mitochondria. Electron microscopy analyses confirmed that treatment of HAECs with IFN-γ or Ad-Bik significantly increased ER-mitochondrial contact. Compared to untreated controls, treatment of cells with IFN-γ for 16 h shortened the mean ER-mitochondria distance from $35.42 \pm 1.37$ to $18.62 \pm 0.85$ and increased the percentage of mitochondria that are in close contact with the ER by >3-fold (Fig. 4c, Table 1). Similar results were observed when cells were infected with Ad-Bik compared with cells infected with Ad-Bik$^{L61G}$ (Fig. 4d, Table 1). Because DAPk1 is a large protein and was essential for Bik-induced Bak co-localization to the ER and increased ER $[Ca^{2+}]_I$ efflux, we hypothesized that DAPk1 may be involved in ER–mitochondria interaction to facilitate the transfer of death signals. HAECs stably expressing shDAPk1 or shCtr treated with IFN-γ or expressing HA-Bik or HA-Bik$^{L61G}$ were immunostained for mitochondria and ER using anti–Cox IV and anti-calnexin antibodies, respectively. Images separated by 0.2 µm to span a total of 10 µm were acquired within 1.1 s to minimize reconstruction artifacts caused by the movement of mitochondria and/or ER. Rotation of the reconstruction on the $y$-axis corroborated that areas of overlap represent juxtaposition of organelles. Areas of ER-mitochondria contact were increased by Bik expression (Fig. 4e) or IFN-γ treatment (Supplementary Fig. 4f) compared to the respective controls. However, the IFN-γ- or Ad-Bik-induced remodeling of ER-mitochondria contact was disturbed in shDAPk1 cells but not in shCtr cells. Also, deficiency of Bak diminished Ad-Bik-induced ER-mitochondrial contact (Fig. 4f), suggesting that ER-Bak anchors DAPk1 to the ER to ultimately tether ER and mitochondria.

**Bik BH3 domain alone is sufficient to enrich for ER-Bak.** To determine whether Bik expression in the airways reduces allergen- or CS-induced ECH and mucous cell metaplasia (MCM), we generated transgenic mice that conditionally induce Bik expression in the respiratory epithelium utilizing the reverse tetracycline transactivator (rtTA) expressed under the control of CCSP promoter. When doxycycline was administered intranasally, the transgene was activated in the CCSP-rtTA/TetOBik mice but not in CCSP-rtTA-only littermates (Fig. 5a, b). Mice were immunized with ovalbumin/Alum and exposed to ovalbumin (OVA) aerosols for 5 days and on the following 2 days instilled with doxycycline. ECH (Fig. 5c) and MCM (Fig. 5d) were significantly reduced in CCSP-rtTA/TetOBik compared to CCSP-rtTA controls. Similarly, when mice were exposed to CS for 3 weeks and instilled with Dox for two consecutive days, ECH (Fig. 5e) and MCM (Fig. 5f) were reduced significantly in CCSP-rtTA/TetOBik compared to CCSP-rtTA controls. Increased transgenic expression of Bik in the airways also caused significant increases in TUNEL positivity compared to controls (Fig. 5g), suggesting that targeted expression of Bik is sufficient to reduce allergen-or CS-induced mucous cells in models of asthma and chronic bronchitis by inducing cell death of hyperplastic cells.

Hydrocarbon-stapled peptide, referred to as stabilized alpha-helix of Bcl-2 domains (SAHBs) were previously described to stabilize the helical conformation and to improve entry into

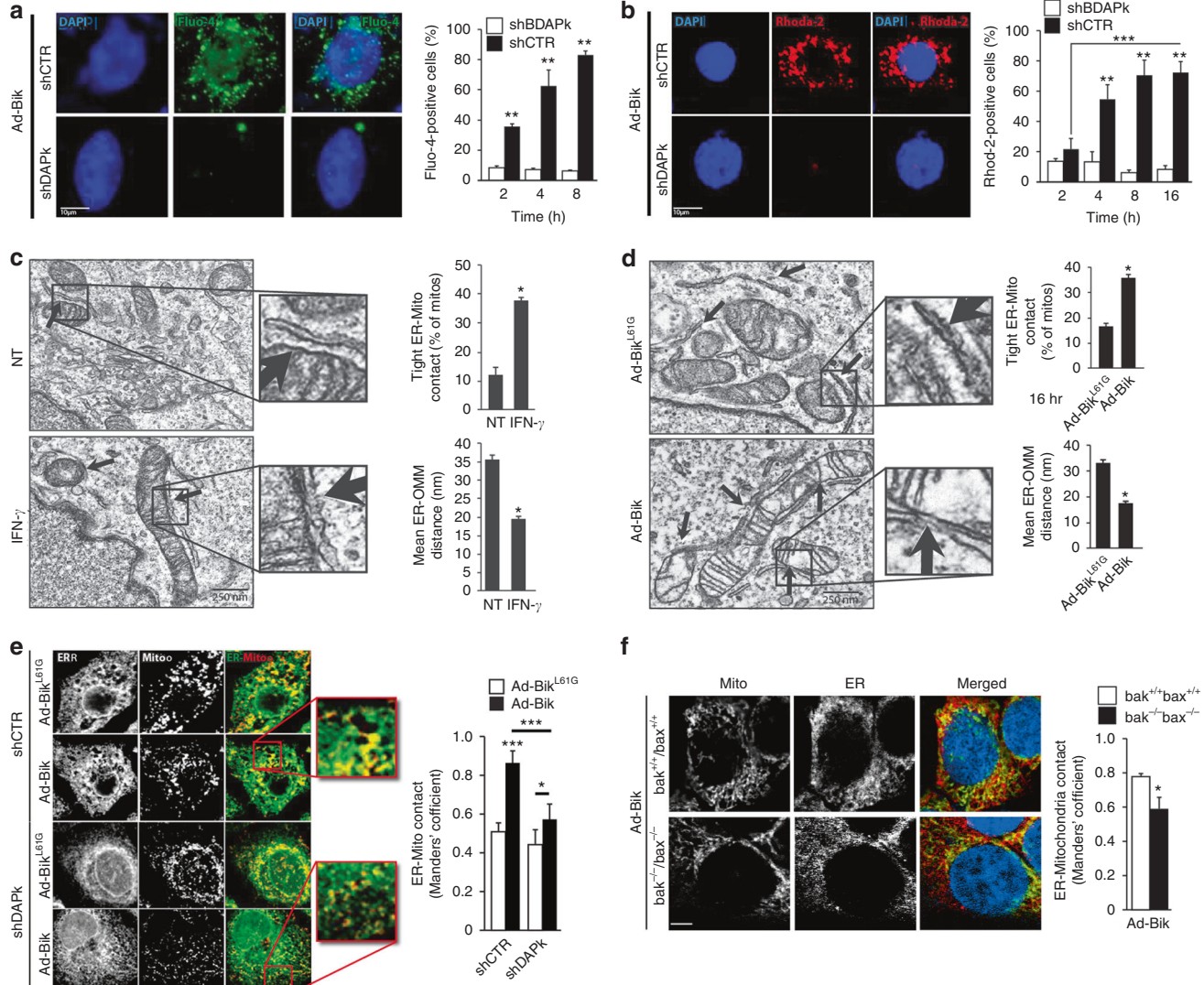

**Fig. 4** Bik facilitates ER Ca$^{2+}$ release and enhances through DAPk1 the proximity of ER and mitochondria. HAECs stably expressing shCtr or shDAPk1 were infected with Ad-Bik and at the indicated time points fixed and stained with Ca$^{2+}$-flux indicator, Fluo-4 (green) (**a**) or with [Ca$^{2+}$]$_m$ indicator, Rhod-2 (red) (**b**) and counterstained for nuclei with DAPI (blue). The percentage of cells positive for Fluo-4 or Rhod-2 were quantified. Graphs show mean ± SEM of quantified cells. **c**, **d** Transmission electron microscopy of HAECs treated with medium alone or 50 ng/ml IFN-γ (**c**) or infected with 100 MOI Ad-Bik or Ad-Bik$^{L61G}$ (**d**) for 16 h. The bar graphs show the mean ER-OMM distances and the percentage of mitochondria with tight contact with the ER. White arrows show contact sites of ER and mitochondria. **e** Representative micrographs of HAECs stably expressing shCtr and shDAPk1, and infected with 100 MOI Ad-Bik or Ad-Bik$^{L61G}$. Cells were stained with anti-CoxIV for mitochondria (green), with anti-calnexin for ER (red), and with DAPI for nuclei staining (blue). **f** bak$^{+/+}$/bax$^{+/+}$ and bak$^{-/-}$/bax$^{-/-}$ HCT116 cell lines were infected with 100 MOI Ad-Bik and 16 h later cells were stained with anti-CoxIV for mitochondria (green), with anti-calnexin for ER (red), and with DAPI for nuclei staining (blue). Confocal images of 3D reconstructions of ER and mitochondria were acquired. Scale bars, 5 μm. ER-mitochondria contacts were quantified by Manders' coefficient, means and SEM (n = 5, at least 40 cells per experiment) of the morphometric data. Differences between two groups were assessed for significance by Student's t test. ANOVA was used to perform pair-wise comparison of the data from more than two groups followed by Fisher least significant difference test. Error bars indicate ± SEM, (n = 5, with > 200 cells analyzed per condition); * = P < 0.05, ** = P < 0.01

cells[34]. To explore the therapeutic potentials of Bik BH3 peptides, we generated a DHS Bik$^{WT}$ BH3 helix (aa 57–71) and a control peptide with the conserved leucine within the BH3 domain mutated to glycine (Bik$^{L61G}$). Four peptides with either a single staple (substitution of Ser63 and Glu67 to amino acids to (S)-2-(2′-pentenyl) Ala residues) or double staple (substitution of Asp55, Leu59, Glu67, and Ser71) and the respective controls with mutation of Leu61 to Gly were synthesized. In addition, one peptide with scrambled amino acid sequence was prepared as further control (Supplementary Fig. 5a). The peptides were labeled with the fluorescent tag carboxyfluorescein (FAM) to monitor uptake into cells. We screened these peptides for cytotoxicity in four airway epithelial cell lines (AALEB, N3, H1975, and HBEC2 cells) (Supplementary Fig. 5a). We found that the (DHS) Bik$^{WT}$ peptides were most effective compared to the scrambled and Bik$^{L61G}$ peptide controls. Treatment of cells with 5 μM of Bik$^{WT}$ peptides was sufficient to cause similar cytotoxicity to 100 MOI Ad-Bik in AALEB cells (Supplementary Fig. 5b) and in primary human AECs from five donors (Supplementary Fig. 5c). The human AECs showed similar uptake of peptides when treated with 5 μM DHS Bik$^{WT}$ or Bik$^{L61G}$ BH3 peptides (Fig. 6a, Supplementary Fig. 5d). Similar to

**Table 1 Measurements of ER-mitochondria interface**

|  | ER-mito distance (nm) | Contacts ≤ 15 nm (%) |
|---|---|---|
| Untreated | 35.42 ± 1.32 ($n = 132$) | 12.26 ± 2.47 ($n = 132$) |
| IFN-γ | 18.62 ± 0.85 ($n = 128$) | 37.54 ± 1.1 ($n = 163$) |
| Ad-Bik$^{L61G}$ | 37.42 ± 1.2 ($n = 163$) | 16.3 ± 1.6 ($n = 163$) |
| Ad-Bik | 19.93 ± 0.69 ($n = 161$) | 35.77 ± 1.34 ($n = 161$) |

the endogenous Bik protein FAM- labeled DHS Bik$^{WT}$ compared to DHS Bik$^{L61G}$ peptide increased Bak protein levels (Fig. 6b), activated Bak (Supplementary Fig. 5e), diminished interaction of Bak and Bcl-2 (Fig. 6c), and impaired cell viability (Fig. 6d). While wild-type and bik$^{-/-}$ MAECs were sensitive to FAM-Bik$^{WT}$ treatment, bak$^{-/-}$ MAECs were protected from FAM-Bik$^{WT}$-induced cell death (Fig. 6e).

The therapeutic role of FAM-Bik$^{WT}$ peptide in reducing allergen- or CS-induced hyperplastic epithelial and mucous cells was tested in mouse models of disease. Sensitized mice that were exposed to OVA aerosols for 5 days were intranasally instilled with 5 μM FAM-Bik$^{WT}$ or FAM-Bik$^{L61G}$ peptides on two consecutive days. Allergen-induced ECH and MCM (Fig. 6f) were reduced significantly in the lung tissues of mice instilled with FAM-Bik$^{WT}$ compared to FAM-Bik$^{L61G}$. In addition, co-localization of ER and Bak were detected in 7% of cells when mice were treated with FAM-Bik$^{WT}$ and in 2.5% in mice treated with FAM-Bik$^{L61G}$ peptides (Fig. 6g). TUNEL positivity was significantly increased in the airways of mice instilled with FAM-Bik$^{WT}$ compared to FAM-Bik$^{L61G}$ (Fig. 6h). Importantly, TUNEL positivity was restricted to the airways. FAM-Bik$^{WT}$ showed similar results in mice exposed to CS for 3 weeks (Supplementary Fig. 5f, g). To test the efficacy of these peptides in human models, we differentiated primary HAECs obtained from five donors in air-liquid-interface cultures over 21 days and treated them with 10 ng/ml IL-13 or 4 μg/ml CS for 48 h. These cultures were then treated with 5 μM FAM-Bik$^{WT}$ or FAM-Bik$^{L61G}$ for 2 days. Application of 5 μM FAM-Bik$^{WT}$ for 48 h caused significant decreases in both IL-13- (Fig. 6i) or CS-induced (Supplementary Fig. 5h) MUC5AC mRNA levels and MUC5AC-expressing cell numbers, compared to FAM-Bik$^{L61G}$ peptides. These findings suggest that small peptides derived from Bik BH3 domain may be used to restore Bik function and reduce allergen- or CS-induced ECH and MCM in asthma and chronic bronchitis, respectively.

## Discussion

The present studies show that the BH3 domain of Bik increases Bak levels to be activated and enriched at the ER. Bik also causes DAPk1 to assemble and form the Bik–DAPk1–ERK–Bak (BDEB) complex. ER Bak anchors DAPk1 at the ER and increases the sites of contact between ER and mitochondria, so that the ER Ca$^{2+}$ that is released due to Bik disrupting the Bcl-2/IP$_3$R interaction is transferred to the mitochondria (Fig. 7). This leads to compromised MOM integrity, release of cytochrome c, and activation of caspase-induced cell death. These studies show that Bik elicits its proapoptotic function by not only activating Bak but also by causing DAPk1 to assemble the BDEB complex and by binding to the anti-apoptotic Bcl-2 to facilitate ER-Ca$^{2+}$ release[35–37].

Although Bik is widely expressed within cells of the hematopoietic compartment and in endothelial cells of the venous lineages, loss of Bik does not increase the numbers of such cells in mice or protect hematopoietic cells in vitro from apoptosis induced by cytokine withdrawal or other cytotoxic stimuli[33]. In mouse embryo fibroblasts, Bik overexpression causes only 10% cell death[38] but up to 60% in AECs[7, 30, 39]. In other cell types,

such as melanoma cells, Bik expression causes DNA fragmentation and chromatin condensation in the absence of caspase activation and cytochrome c release[40]. This cell-type-specific effect of Bik may be due the presence/absence of proteins that ultimately form the BDEB complex. We found that cells expressing low levels of DAPk1 or Bak are less sensitive to Bik, suggesting that abundant expression of DAPk1 and Bak activation in the ER are crucial for Bik to initiate cell death. Overexpression of GRP78 counteracts the Bik-mediated cell death in 293T cells, while Bik-induced cell death is enhanced when GRP78 levels are suppressed[41]. Therefore, at baseline, low levels of Bik may be inhibited by GRP78 from forming the BDEB complex and increased Bik expression by IFN-γ or adenoviral expression vector may be sufficient to provide excess Bik that activates Bak and enriches for ER Bak.

Our earlier studies found that IFN-γ induced AEC death involves the ER and found that bax$^{+/+}$ and bax$^{-/-}$ mouse AECs showed minor differences in IFN-γ-induced cell death[24]. However, when investigating the role of Bak using bak$^{+/-}$ and bak$^{-/-}$ MAECs, the protection by reduced Bak levels were far greater than by deficiency in Bax. Confirmation that Bak independent of Bax is crucial for IFN-γ- and Bik-induced cell death pathway was demonstrated by restoring Bak$^{WT}$ or Bak$^{Y108A}$ in HCT116$^{bax−/−bak−/−}$ cells. Bak is a phospho-protein and its activation involves a series of conformational changes, including exposure of the occluded N-terminal epitopes[27, 42] followed by the formation of homo-oligomeric complexes that permeabilize the MOM[43, 44]. Such conformational changes and oligomerization require dephosphorylation of Bak at Tyr108 (Y108) and Ser117[28]. Further, when Bak undergoes the N-terminal conformational change, the BH3 domain is exposed[45] that may promote the oligomerization of reticular Bak[43]. Interestingly, Bik expression was required to elicit Bak oligomerization even when Bak$^{Y108A}$ was expressed, suggesting that inactivation of Bcl-2 may be a crucial trigger for Bak to form oligomers. The oligomerized Bak anchors DAPk1 to the ER and provides the means for the released ER Ca$^{2+}$ to enter mitochondria that are in close proximity.

The Bik BH3 domain is sufficient for efficient heterodimerization with Bcl-2 but this heterodimerization is not sufficient to cause death in MCF-7 cells[46]. Our studies show that the Bik BH3 domain dissociates Bak from the Bcl-2/Bak complex, activates and increases the ER-localized Bak levels by > 5-fold. That Bik dissociates Bak from Bcl-2 suggests that the BH3 domain of Bik has a higher affinity to Bcl-2 than to Bak. More interestingly, the dissociation of Bak from Bcl-2 led to stabilization and increase in Bak levels, indicating that the heterocomplex, Bak/Bcl-2, is degraded by the proteasomal pathway at a higher rate than the Bak monomer. Further, while blocking proteasomal degradation with MG-132 increases Bak levels, suppression of Bcl-2 expression further enhances stability of Bak, suggesting that Bcl-2 may promote Bak degradation by mechanisms that do not involve the proteasome. The underlying mechanisms for this difference in degradation need further investigation.

The Bik BH3 domain also modifies the conformation of DAPk1 to facilitate the binding to the other proteins because mutation of Lys42 that is crucial for Bik interaction with DAPk1 also disrupts the complex formation and cell death. Because the BDEB complex is no longer formed when expression of DAPk1 is suppressed, it is likely that DAPk1 may serve as a scaffold, whereby Bik binds to the kinase domain and activated Bak binds to the kinase and CaM domains of DAPk1. DAPk1 sequesters ERK1/2 in the cytoplasm by interacting with ERK through its death domain to promote the proapoptotic function of DAPk1[31]. Similarly, Bik mediates IFN-γ-induced cell death in the AECs by interacting with and sequestering ERK1/2 in the cytosol[5]. However, deletion of the death domain of DAPk1 did not affect the

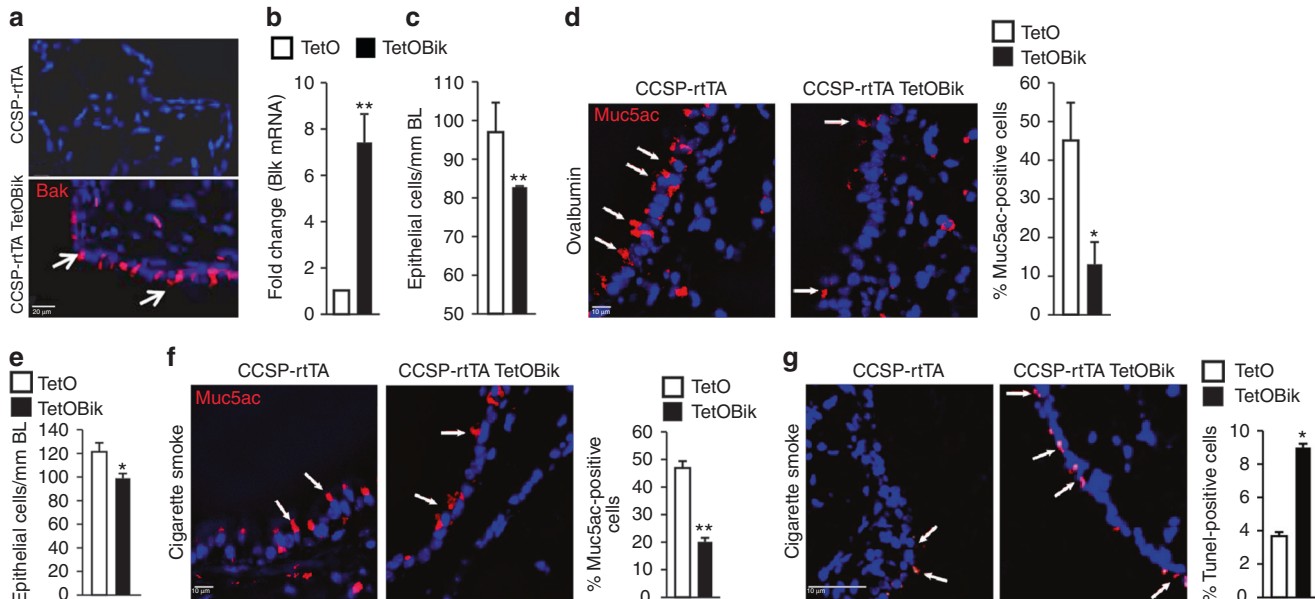

**Fig. 5** Inducible expression of Bik in the airways of adult mice reduces allergen- or CS-induced MCM. CCSP-rtTA+/TetoBik− and CCSP-rtTA+/TetoBik+ mice were immunized with ovalbumin/Alum and exposed to ovalbumin aerosols for 5 days and on the following day instilled with 100 ng doxycycline or PBS intranasally. Lungs were collected 24 h later and analyzed for airway-specific expression of (**a**) Bik protein by immunostaining and (**b**) Bik mRNA expression level by qRT-PCR. Representative micrographs of airways, white arrows indicate Bik-positive cells. **c**, **d** CCSP-rtTA and CCSP-rtTA/TetOBik mice were immunized with OVA/Alum and exposed to OVA aerosols for 5 days and instilled with doxycycline intranasally for 2 days after the last OVA exposure. The number of airway epithelial cells/mm BL (**c**) and the percentage of Muc5ac-postive cells (**d**) were quantified in AB/H&E stained or immunostained paraffin-embedded lung tissues. **e**, **f** Mice exposed to CS for 3 weeks followed by one time intranasal instillation of 100 ng doxycycline or PBS. CS-induced ECH (**e**) and MCM (**f**) were significantly reduced in the airways of CCSP-rtTA/TetOBik compared to CCSP-rtTA mice. White arrows indicate Muc5ac-positive cells. **g** Lung tissues were analyzed for internucleosomal DNA fragmentation using TUNEL assay. White arrow indicates TUNEL-positive cells. Differences between two groups were assessed for significance by Student's *t* test. Error bars indicate ± SEM, (*n* = 8 mice per group); * = P < 0.05; ** = P < 0.01

interaction of DAPk1 with Bik or Bak, suggesting that interactions of DAPk1 with Bik and Bak occurs independent of ERK–DAPk1 interaction. At present, the only evidence for Bik and DAPk1 interaction is based on co-immunoprecipitation of over-expressed Bik and future studies need to validate this finding using other approaches such as fluorescent polarization assays. These studies would confirm the possibility that DAPk1 may have a Bcl-2-like binding groove in the kinase domain.

Disturbances in cellular $Ca^{2+}$ homeostasis, such as cytosolic $Ca^{2+}$ overload, ER $Ca^{2+}$ depletion, and mitochondrial $Ca^{2+}$ overload can lead to increased mitochondrial permeability and cytochrome c release[15]. On the basis of previous studies[11, 12], we have to assume that ER-$Ca^{2+}$ precedes the accumulation of mitochondrial $Ca^{2+}$. Our studies also support the idea that ER-Bak is the main driver of ER-$Ca^{2+}$ release that results in mitochondrial $Ca^{2+}$ accumulation. We found that Bik not only dissociated Bak from Bcl-2 but also disrupted the $IP_3R$–Bcl-2 interaction to cause the opening of the ER calcium channel, $IP_3Rs$[29, 47, 48]. The juxtaposition of ER and mitochondria is required for efficient mitochondrial $Ca^{2+}$ uptake[15]. In a healthy cell, overlapping regions between ER and mitochondria are estimated to cover 5–20% of total mitochondrial surface, enabling microdomains of high $Ca^{2+}$ concentration and fine-tuning $Ca^{2+}$ fluxes between these organelles[49–51]. The extent of ER mitochondria contact, as measured by areas of distances between the ER and mitochondria that are <15 nm apart, increased to 35% in IFN-γ or Bik expressing cells. The finding that $bak^{+/−}$ cells were protected from death suggests that the amount of activated ER Bak is a determinant for extent of ER and mitochondria tethering across the cell. In addition, the Bik BH3 stabilized alpha-helix of Bcl-2 domains (SAHBs)[34] was capable of causing cell death by

dissociating Bcl-2/Bak complex and activating Bak. The fact that the ER anchoring domain of Bik was not required for Bak activation and cell death is further supporting evidence that ER Bak but not Bik is responsible for anchoring DAPk1 to the ER. The resulting frequency of contact sites between ER and mitochondria determine extent of MOMP disruption and cytochrome c release that ultimately activates downstream caspases.

Our studies are the first to identify DAPk1 as an ER-mitochondrial tethering protein to facilitate IFN-γ- and Bik-induced ER $Ca^{2+}$ release and caspase activation. Future studies will address the involvement of proteins known to attach DAPk1 to mitochondria. As a large protein, DAPk1 interacts with numerous proteins, including ZIPK[52]; HSP90, CHIP, and DIP[53, 54]. DIP localizes to the mitochondria to cause caspase-dependent cell death[55], therefore, it is possible that any of these mitochondrial proteins binds to DAPk1 to help tether ER to mitochondria.

The Bik BH3 SAHB peptide was sufficient to trigger cell death in a pathway similar to the whole Bik protein that was expressed in AECs of adult mice using transgenic approach. Intranasal delivery of Bik$^{WT}$, but not Bik$^{L61G}$ peptides, caused cell death restricted to the airways and resolved allergen- or CS-induced MCM and ECH. Bik peptide increases the level of Bak protein by displacing Bak from Bcl-2. Bik peptides also caused Bak to co-localize with the ER and increase TUNEL positivity in the airways of mice in vivo. These findings support the fact that stapled Bik peptides cause cell death in the AECs by the same pathway as the Bik protein and that small peptides derived from Bik BH3 domain may be used to restore Bik function and control allergen- or CS-induced ECH and MCM.

Several studies have shown that secretory cells are the cells that proliferate following injury to the epithelium[56–58]. The cell death

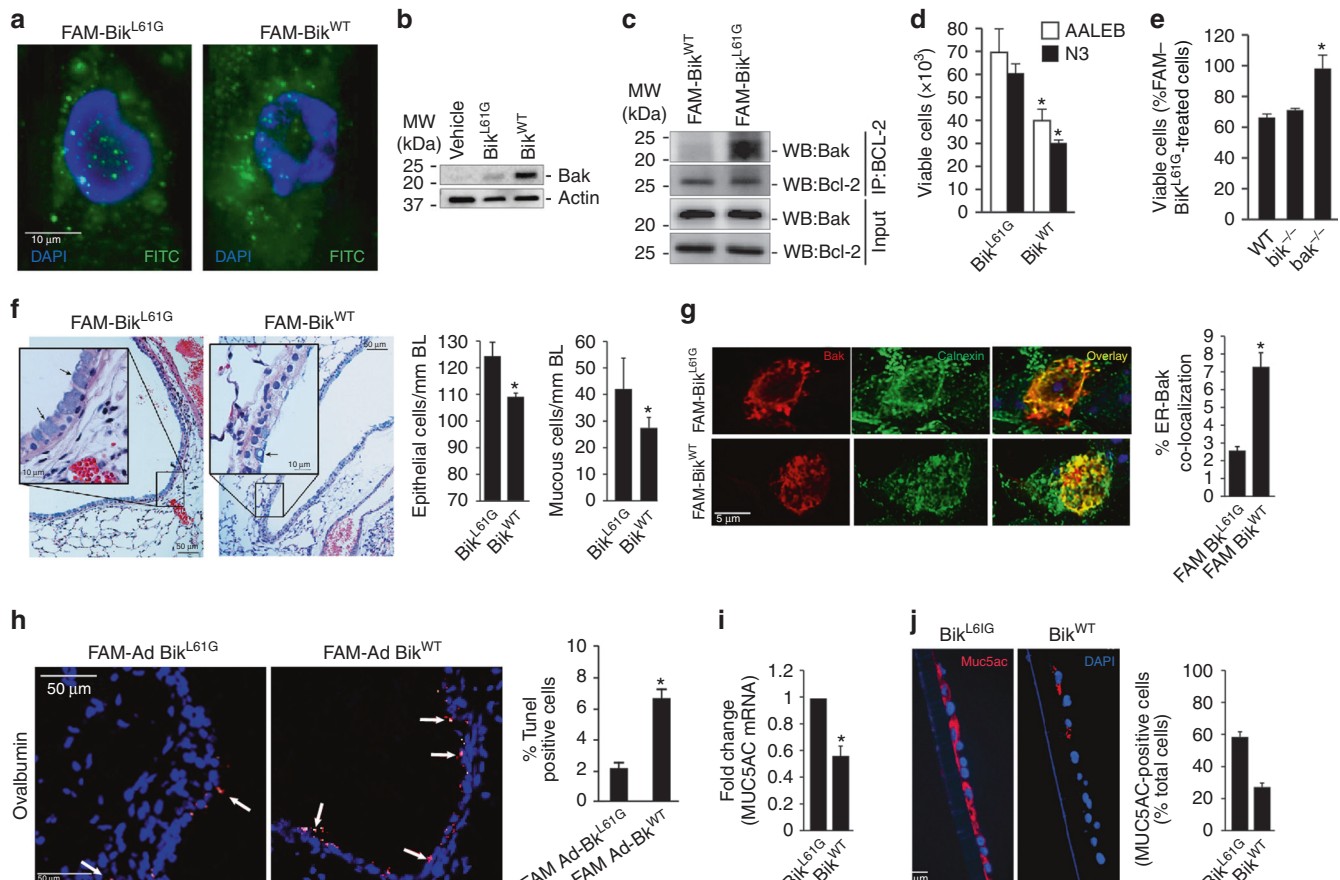

**Fig. 6** Hydrocarbon-stapled Bik BH3 peptide is sufficient to activate Bak and reduce allergen- or CS-induced MCM, and ECH in vivo. **a** Confocal microscopy of HAECs treated with FAM-Bik$^{WT}$ or FAM-Bik$^{L61G}$ (1 μM, 2 h) demonstrate sufficient uptake of peptides. **b** Protein lysates from HAECs treated with vehicle only, or 5 μM of FAM-Bik$^{WT}$ or FAM-Bik$^{L61G}$ were analyzed for Bak expression by western blotting. **c** Protein lysates from HAECs treated with FAM-Bik$^{WT}$ or FAM-Bik$^{L61G}$ were immunoprecipitated with anti-Bcl-2 antibody and analyzed for Bak levels by western blotting. **d** HAEC cell lines (AALEBs and N3) were treated with 5 μM of FAM-Bik$^{WT}$ or FAM-Bik$^{L61G}$ peptides for 72 h and analyzed for cytotoxicity by trypan blue exclusion assay. **e** MAECs from wild-type, $bik^{-/-}$, or $bak^{-/-}$ mice were treated with 5 μM FAM-Bik$^{WT}$ or FAM-Bik$^{L61G}$ peptides and viable cells were quantified. **f** C57BL/6 mice were immunized with 1 μg OVA/Alum on days 1 and 7, and were exposed to 2 mg/m$^3$ OVA aerosols for 5 d. Mice were then instilled with FAM-Bik$^{WT}$ or FAM-Bik$^{L61G}$ in 50 μL PBS on 2 consecutive days and lungs were collected the following day. AB/H&E staining of paraffin-embedded lung tissues show that ECH and MCM were significantly reduced in the airways of mice instilled with FAM-Bik$^{WT}$ compared to FAM-Bik$^{L61G}$. **g** Paraffin embedded lung tissues from mice exposed to ovalbumin and instilled with 5 μM FAM-Bik$^{WT}$ or FAM-Bik$^{L61G}$ for 48 h were stained with anti-calnexin and anti-Bak antibodies, and the percentage of cells in the airways with Bak co-localized to the ER were compared after confocal microscopy. **h** Lung tissues were analyzed for internucleosomal DNA fragmentation using TUNEL assay. White arrow indicates TUNEL-positive cells. **i** Differentiated primary HAECs from 5 donors were treated with 10 ng/ml IL13 for 72 h followed by treatments with 5 μM FAM-Bik$^{WT}$ or FAM-Bik$^{L61G}$ for 48 h. MUC5AC mRNA levels were analyzed by qRT-PCR. **j** Paraffin-embedded cells were immunostained for MUC5AC positivity using α-MUC5AC antibody and DAPI. The data presented are means ± SEM for two independent experiments conducted in triplicates. Differences between two groups were assessed for significance by Student's $t$ test. ANOVA was used to perform pair-wise comparison of the data from more than two groups followed by Fisher least significant difference test. Error bars indicate ± SEM, ($n = 8$ mice per group); * = $P < 0.05$

caused by Bik BH3 peptides in vivo may be restricted to hyperplastic mucous cells of the airways as most of the epithelial cells remain unharmed even when Bik is expressed. This removal process may also occur rapidly, supporting the observation that the percentage of TUNEL-positive cells and cells with ER-Bak detected in the airways of transgenic mice or mice treated with DHS Bik$^{WT}$ at a single time point is below 10%. We have not observed any evidence of gaps in the basement membrane or damage to epithelium in $bak^{+/+}$ or $bak^{-/-}$ mice when the resolution of ECH occurs over a period of time. Detailed mechanisms of the removal process of dying cells from the epithelium will be studied in the future. Targeted reduction of hyperplastic mucous cells by restoring the normal resolution and recovery process is of advantage because it does not interfere with the normally important innate immunity role of mucus synthesis and

secretion. Therefore, these studies potentially may lay the foundation for better therapies to reduce ECHs in general or hyperplastic mucous cells in particular without destroying the integrity of the airway epithelium and its innate protective mechanism. The finding that Bik increased Bak and caused colon goblet cell apoptosis suggests that this approach may be useful for treating diseases associated with ECH other than in airways.

## Methods

**Mice**. Male-specific pathogen-free wild-type C57BL/6 and $bak^{-/-}$ mice on C57BL/6 background were purchased from The Jackson Laboratory (Bar Harbor, ME). Mice were housed in isolated cages under specific pathogen-free conditions. After a 14-day quarantine period, mice were acclimatized for 8 days and entered into the experimental protocol at 8–10 weeks of age. The $bik^{+/-}$ mice on C57BL/6 background were provided by Andreas Strasser (Walter and Eliza Hall Institute, Melbourne, Australia). $bik^{+/+}$ with $bik^{-/-}$ littermates were bred from the respective

**Fig. 7** A schematic depicting that Bik fulfills several functions to facilitate cell death. (1) Bik dissociates Bak/Bcl-2 and interacts with DAPk1 to form a complex with Bak that forms multimers and anchors DAPk1 to the ER to increase the contact sites of ER and mitochondria, (2) Bik disrupts the interaction between Bcl-2 and $IP_3R$ to cause calcium release, and (3) ER-associated Bak interacts with DAPk1 to facilitate mitochondrial calcium uptake

heterozygote mice at the Lovelace Respiratory Research Institute under specific pathogen-free conditions and genotyped[33]. Transgenic mice that conditionally express Bik in the respiratory epithelium were generated in our lab utilizing the reverse tetracycline-controlled transactivator (rtTA) expressed under the control of CCSP promoter[59]. Mice were exposed to 250 mg/m$^3$ CS or filtered air for 6 h per day, 5 days per week for 3 weeks or were allowed to recover in air for an additional 8 weeks after 3 weeks of CS exposure. Sensitization and exposure of mice to OVA were performed[30, 60]. After 5 days of exposure to OVA or 3 weeks of exposure to CS, mice were anesthetized with isoflurane and intranasally instilled with FAM-Bik$^{WT}$ or FAM-Bik$^{L61G}$ as a control in a volume of 50 ml PBS on days 1 and 2 after the last day of exposure. On day 3, eight mice from each group were killed and right lung tissue was collected and used to analyze the mRNA expression levels of Bak and Muc5ac using qRT-PCR. Preparation of lung tissues for histopathological examination was performed[3]. Briefly, left lungs were inflated and fixed at 25 mm pressure with zinc formalin for preparing tissue sections and evaluating ECH and MCM. Tissue sections were stained with Alcian blue (AB) and periodic acid Schiff or hematoxylin and eosin[61]. The number of AB-positive cells and per total cell counts per millimeter of BL were quantified using the VisioMorph system (Visiopham, Horsholm, Denmark). All experiments were approved by the Lovelace Institutional Animal Care and Use Committee and were conducted at Lovelace Respiratory Research Institute, a facility approved by the Association for the Assessment and Accreditation for Laboratory Animal Care International.

**Cells**. MAECs were isolated from the trachea of 6 to 8-weeks-old C57BL/6 mice and cultured on plastic plate or Transwell membranes (Corning, New York, NY) after seeding with $4 \times 10^4$ to $9 \times 10^4$ cells[62]. Primary HAECs were purchased from Cambrex Bio Science Walkersville, Inc (Walkersville, MD). Primary murine colon epithelial cells were isolated from 4-weeks-old C57BL/6 mice[63]. Briefly, intestinal contents were cleaned and colons were flushed with cold PBS. The colon was longitudinally opened and washed with ice cold calcium- and magnesium-free HBSS (Gibco), supplemented with 100 mg/ml streptomycin, 100 U/ml penicillin, 0.5% gentamicin, and 1% fungizone (Gibco). After colons were cut into small pieces and washed with HBSS buffer, they were partially digested with 0.2% collagenase I (Sigma-Aldrich) and 1 mg/ml dispase in Dulbecco's Modified Eagle's medium (DMEM) media supplemented with 2% Luria broth, glutamine, penicillin, streptomycin, gentamycin and fungizone, and 10% fetal bovine serum at 37 °C on a shaker for 120 min. The digested tissues were subjected to four washes with HBSS and plated on 6-well plates and dissociated cells were cultured in DMEM medium supplemented with 10% FBS. To remove fibroblasts, cells were transferred to fresh plates every 4 h for 3 consecutive times. After adding fresh media, cells were allowed to grow for 2–3 days.

The immortalized HAECs, AALEB and N3 cells, provided by S. Randell (University of North Carolina Chapel Hill, Chapel Hill, NC)[64, 65]. Wild-type and HCT116$^{bax-/-bak-/-}$ double knockout cells were kindly provided by Dr. Richard Youle, National Institute of Neurological Disorders and Stroke (NINDS), were cultured in McCoy's Media supplemented with 10% FBS and glutamine. H1975, HEK293T, Calu-6, H4006, H23, T47D, and H1299 cells were purchased from ATCC and cultured in RPMI media supplemented with 10% FBS and glutamine. HEK293 and MCF-7 cell lines were obtained from ATCC and cultured in DMEM media supplemented with 10% FBS and glutamine. H2009 cell line was obtained from ATCC and cultured in F12 (50%) and DMEM (50%) media supplemented with 10% FBS and glutamine. Cells were transfected with plasmid DNA using TransIT-2020 transfection reagent (Mirus, Madison, WI). Cells were tested for mycoplasma contamination.

**Plasmids, adenoviral constructs, and reagents**. Adenoviral expression vectors for Bik and Bik$^{L61G}$ were provided by G. Shore (McGill University, Montreal, Quebec, Canada)[66]. Flag-tagged DAPk1 and DAPk1$^{K42A}$ were provided by Dr. Youming Lu. (Biomedical Sciences Center, University of Central Florida, Orlando, FL) GFP-tagged deletion constructs of DAPK domains were generated using standard molecular biology techniques. Primers specific for DAPk1 were used to

amplify the sequence coding for amino acid 10274, 275–650, 651–900, 901–1250, 1251–1430 by PCR. Primers contain restriction sites for ECOR1 on the 5′ side and for NOT1 on the 3′ side. PCR products were digested with EcoRI and NotI, gel purified and inserted downstream of the triple flag-tagged ypet in the pcDβA F3 ypet plasmid so that expression of the fusion protein is under the control of the CMV enhancer. Bak constructs (pcDβA F3CypetR (G4S)4-F3hBak, hBak-wt (1–183) F3CypetR, or F3YpetR-F3hBak) were generated by PCR and standard cloning procedures. Briefly, the sequence of wild-type Bak was amplified with specific primers flanked by EcoRI and Xhol restriction sites to allow the cloning into a mammalian expression vector in a frame with triple FLAG-tagged Cypet (G4S)$_4$. The resulting chimeric Bak protein and the Cypet were separated by 51 aa consisting of the (G4S)$_4$ Flag3 sequences. The other two Bak constructs were generated in a similar manner. Plasmid expression vector for Bcl-2-5b were provided by Dr. David Andrews (Departments of Biochemistry and Medical Biophysics, University of Toronto). Pan-caspase inhibitor Q-VD-Oph was purchased from R&D Systems, Inc. and was used in selected experiments.

Because all of our studies showed that the BH-3 domain is the active site, we designed several peptides that comprise the BH-3 domain. Previous studies have demonstrated that modification of these peptides by formation of hydrocarbonstaples, referred to as stabilized alpha-helix of Bcl-2 domains (SAHBs), stabilizes helical conformation and increases stability[34] for Bim[67], Bid[68], and Bad[69] BH3 domains. We designed the substitution of amino acids to (S)-2-(2′-pentenyl) Ala residues based on 3-D structures for Bad[69]. Peptides were also labeled with the fluorescent tag carboxyfluorescein (FAM) to better monitor uptake into cells. Bik-derived peptides with single or double staples and the mutant controls (Leu changed to Gly) were synthesized to demonstrate that a single amino acid substitution is sufficient to discriminate in the killing activity. In addition, a peptide with scrambled amino acid sequence was prepared as further control. The high-performance liquid chromatography (HPLC) and mass spectrometry (MS) data for the peptides have been included as Supplementary Figs. 12–16.

**Retroviral silencing**. Retroviral silencing vector encoding for Bik shRNA, DAPk1 and the control vector were purchased from Origene (Origene Technologies, Inc). The suppressing effect of the shRNA was established in HAECs, and amplification and purification of plasmid DNA and packaging of the retroviral particles were performed in Phoenix cells as specified by the manufacturer's instructions. Briefly, Phoenix packaging cells were transfected with retroviral constructs and supernatants containing packaged virus particles were collected during the 48–96 h period after transfection, centrifuged at 2000 rpm for 10 min to remove packaging cells, and stored at −80 °C in aliquots. HAECs were infected with viral supernatant in the presence of 10 µg/ml of polybrene (Sigma-Aldrich). Stable cell lines expressing Bik, DAPk1 and control shRNAs were generated by selecting infected cells with 1 µg/ml puromycin (Calbiochem Inc.).

**Subcellular fractionation**. IFN-γ-treated or Ad-Bik- or Ad-Bik$^{L61G}$-infected cells were collected at 10, 24 or 48 h and incubated for 10 min on ice with hypotonic buffer (10 mM HEPES pH 7.8, 1 mM EGTA, 1 mM KCL and protease inhibitors 1 mM PMSF, 80 µM aprotenin, 4 mM bestatin, E-64 1.4 mM, 2 mM leupeptin and 1.5 mM pepstatin A). Isotonic buffer (10 mM HEPES pH 7.8, 250 mM sucrose, 1 mM EGTA and 2 mM KCl) was added before lysing the cytoplasmic membrane by dounce homogenization. Following a centrifugation at $3000 \times g$ for 10 min to remove the nuclei and heavy membranes (mitochondrial fraction), the supernatant was further centrifuged at $100,000 \times g$ to precipitate the light membranes that represent the ER and obtain the cytosolic fractions as the supernatant. Approximately $5 \times 10^7$ cells were used for each time point to obtain sufficient protein material for analyzing the components by western blotting. Cytosolic and nuclear fractions were prepared by lysing cells in NP-40 to obtain the cytosolic fraction and extracting the nuclear proteins with a hypertonic extraction buffer (50 mM N-2-hydroxyethylpiperazine-N9-ethane sulfonic acid, pH 7.8, 50 mM KCl, and 300 mM NaCl) in the presence of protease and phosphatase inhibitors[70].

**Protein cross-linking**. Mitochondrial or ER fractions were incubated with the 5 mM cross-linking agent, BMH (Pierce Chemical Co.) for 30 min at room temperature. After the reaction was stopped by adding 5 mM DTT for 15 min, samples were diluted with SDS sample buffer, heated to 95 °C for 5 min, separated on 12% (wt/vol) SDS-PAGE, transferred to nitrocellulose, and probed with anti-Bak antibodies.

**Immunoprecipitation**. Immunoprecipitation from protein preparations were performed using the Pierce Crosslink Immunoprecipitation Kit (Pierce Biotechnology, Rockford, IL) according to the instruction of manufacturer. Briefly, cell lysates were incubated with antibody-cross-linked Pierce Protein A/G Plus Agaros overnight at 4 °C. The immunoprecipitates were eluted and subjected to Western blot analysis.

**Detection of intracellular Ca²⁺**. Cells were washed three times with HBSS medium (Life Technologies Inc., Grand Island, New York) and incubated with 5 μM Fluo-4 AM (Life Technologies Inc., Eugene, OR) or 10 μM Rhod-2 AM (Life Technologies Inc., Eugene, OR) as an indicator of $Ca^{2+}$ in the cytosol ($[Ca^{2+}]i$) or mitochondria ($[Ca^{2+}]m$), respectively, at 37 °C for 45 min. Cells were washed three times with HBSS, fixed with 3% paraformaldehyde, mounted with DAPI-containing Fluormount-G (SouthernBiotech, Birmingham, AL) for nuclear staining, and analyzed with a fluorescence microscope.

For calcium release assays using fluorimeter, intracellular calcium was detected by staining cells with Fluo-4 dye (Invitrogen) and a single-excitation (490 nm) fluorophore for which emission intensity (510 nm) that is directly proportional to the level of bound calcium. Alternatively, an acetoxymethyl ester (AM) derivative of Fluo-4 was used for staining cells as the derivative can cross the plasma membrane and following the cleavage of AM moiety by endogenous esterases the charged dye remains trapped within the cell. Fluorescence was measured at excitation/emission of 485/535 using a Fluoroskan Ascent plate reader (Labsystems) and the relative fluorescent units (RFU) were calculated.

**Quantitative RT-PCR**. For quantitative RT-PCR, the primer/probe sets (Applied Biosystems) were distributed into each well in duplicates, and target mRNAs were amplified by PCR in 20-μl reactions. Preamplification efficiency was assessed by performing amplification of nonamplifed cDNA with TaqMan Gene Expression Assays (Applied Biosystems). For all reactions, CT values >37 were eliminated for evaluation of preamplification efficiency. Uniform preamplification was demonstrated by a ΔΔCT value of −1.5 to 1.5 when comparing the CT values of each gene amplified from preamplified and nonamplified cDNA. Because all results were derived from the linear amplification curve, the use of the ΔΔCT method ensures that only mRNA amplification within the linear range was compared.

**Western blot analysis**. Protein lysates for western blot were prepared by lysing cells in radioimmunoprecipitation assay (RIPA) buffer (10 mM Tris, pH 7.4, 150 mM NaCl, 1% Triton X-100, 1% deoxycholate, 0.1% SDS, 5 mM EDTA) supplemented with the protease inhibitors PMSF (1 mM), pepstatin (10 μg/ml), aprotinin (2 μg/ml), and benzamidine (2 μg/ml). Protein concentration was determined by BCA kit (Pierce; Rockford, IL). A total of 40–8 μg protein was electrophresed on SDS-PAGE gel, blotted to PVDF membrane, and sequentially probed with primary antibodies. A horseradish peroxidase-conjugated secondary antibodies were used and secondary antibodies were detected using enhanced chemiluminescence (Plus-ECL, PerkinElmer.Inc., MA)[60]. The following antibodies were used: Rabbit anti-Bik (Cat.#4592), rabbit anti-Cox IV (Cat.#4844), rabbit anti–phospho-ERK1/2 (cat. #9101), and rabbit anti-ERK1/2 antibodies (Cat.#9102), rabbit anti-pSTAT1, rabbit anti-Bcl-2 antibody (Cat.#4223S) were obtained from Cell Signaling Technologies (Cell Signaling Technologies, Danvers, MA) and used at 1:1000 dilutions. Mouse anti-Bak (Cat.#ab104124), rabbit anti-calnexin (Cat.#ab22595), mouse Cytochrome C (Cat.#ab13575) antibodies were obtained from Abcam Inc. (Cambridge, MA) and used at 1:500 dilutions. Mouse anti-HA (cat.#MSA-106) was from Stressgen (Ann Arbor, MI) and used at 1:1000 dilution, mouse anti-active Bak antibody (cat. #AMO3) from Calbiochem (EMD Chemicals Inc., Darmstadt, Germany) and used at 1:500 dilution, mouse anti-DAPk1 antibody (Cat.#610290) from BD Biosciences (San Jose, CA) and used at 1:500 dilution, rabbit anti-GFP antibody (Cat.#3999-100) from BioVision Inc. (Milpitas, CA) and used at 1:250 dilution, mouse anti-Flag antibody (Cat.#F3165) from Sigma-Aldrich Inc. (St. Louis, MO) was used at 10 μg/ml concentration, rabbit anti-IP3R antibody (Cat.#ABS2129) from Calbiochem (EMD Millipore Corporation, Billerica, MA) and used at 1:1000 dilution. Uncropped scans of blots are provided as supplementary figures in the Supplementary Figs. 6–11.

**Immunofluorescence**. For cytometry, cells were grown on Lab-Tek-II eight-chamber slides (Nalge Nunc International, Rochester, NY) and after treatments were fixed using 3% paraformaldehyde with 3% sucrose in PBS and processed for immunostaining. Briefly, after antigen retrieval, sections were incubated with a 1:500 dilution of anti-Bak antibodies at 4 °C overnight. For mitochondrial and ER co-localization studies, cells were incubated with either Cox IV (Cell Signaling Technologies, Danvers, MA) for 30 min or anti-calnexin (Abcam Inc., Cambridge, MA) antibodies overnight. The immunolabeled cells were detected using F(ab′)₂

fragments of respective secondary Abs conjugated to either Dylight-549 or Dylight-649 (Jackson ImmunoResearch Laboratories, West Grove, PA) at 1:1000 dilution and mounted with DAPI-containing Fluormount-G (SouthernBiotech, Birmingham, AL) for nuclear staining Immunofluorescence was imaged using Axioplan 2 microscope (Carl Zeiss, Inc., Thornwood, NY) with a Plan-Neofluor 403/0.75 air objective and a charge-coupled device camera (Hamamatsu Photonics, Hamamatsu, Japan) with the acquisition software Slidebook 6.0 (Intelligent Imaging Innovation, Denver, CO).

**Flow cytometry**. To analyze for Bak activation, cells were trypsinized, washed with PBS, fixed in 0.25% paraformaldehyde, and permeabilized with 0.01% saponin/PBS before the incubation with anti-human Bak monoclonal primary antibody (Clone Ab-1, AM03; Calbiochem) for 30 min at 4 °C. Cells were then washed and incubated with Alexa647-labeled goat anti-mouse (Invitrogen Inc.) antibody for 30 min at 4 °C. Cells were washed and resuspended in 0.2% BSA/PBS for analysis with a BD FACSCanto® Flow Cytometer (BD Biosciences). Fluorescence was acquired using logarithmic amplifiers. Approximately, 10,000 cells were analyzed per sample. Flow cytometric results were quantified by manipulating the raw data as described (Griffiths, 1999) using FlowJo analysis software (Treestar Inc). Briefly, cells exhibiting a light scatter profile associated with apoptotic cells were gated out and the median Bak-associated fluorescence was determined by subtracting the median fluorescence of the mouse IgG control from each test sample. The median value was then multiplied by the percentage of Bak-positive cells as determined by the IgG control to give the Ab-1 Bak-specific fluorescence of each sample.

Early and late apoptotic cells were quantified by fixing collected cells in 0.25% paraformaldehyde and incubating them with Annexin V-FITC conjugate (BD Biosciences Inc., San Jose, CA) and propidium iodide (PI, Sigma-Aldrich Inc., St. Louis, MO) for 30 min at 4 °C. After washing, resuspended cells in 0.2% BSA/PBS ~10,000 cells per sample were analyzed using BD FACSCanto® Flow Cytometer (BD Biosciences Inc., San Jose, CA). Flow cytometric results were quantified by manipulating the raw data as described[27] using FlowJo analysis software (Tree Star Inc., Ashland, OR). Detection of Annexin V-positivity (AnnV+/PI−) was used as a marker for early apoptotic cells, propidium iodide-positivity (AnnV−/PI+) was used as necrotic cell marker, the double-positive (AnnV+/PI+) cells represented late apoptotic cells and double-negative (AnnV−/PI−) cells represented viable cells.

**ER-mitochondria co-localization**. The ER and mitochondria were immunostained in fixed cells using rabbit anti-calnexin (Abcam Inc., Cambridge, MA) and mouse anti- cyclo-oxygenase IV antibodies (CoxIV, Cell Signaling Technology Inc., Beverly, MA), respectively, followed by detection with Alexa-546 or -647 -conjugated respective secondary antibodies (Invitrogen). The cells were then mounted with Fluormount-G containing DAPI (for nuclear staining). The confocal stacks were acquired every 0.2 μM along the z-axis (for a total of 20–30 images) with a 100X objective using a Zeiss 510-Meta confocal system (CarlZeiss Inc.). For ER–mitochondria interaction analysis, stacks were automatically thresholded using ImageJ, 3D reconstructed and surface rendered. Interactions were quantified by Manders' colocalization coefficient as described[13] after the background was subtracted using BG subtraction function (ImageJ).

**Transmission electron microscopy**. HAECs were fixed with 2.5% glutaraldehyde, 4% paraformaldehyde in 0.1 M phosphate buffer (pH 7.2) overnight at 4 °C. Cells were subsequently postfixed with 1% osmium tetroxide in PBS, infiltrated with LX-112 resin (Ladd Research, VT), and embedded on the culture dish, an appropriate sized block face trimmed, and 70-nm ultrathin sections cut with a Diatome diamond knife in an MT5000 ultramicrotome (Sorvall Instruments Div., Newton, CT). Sections were picked up on parlodian-coated copper EMi grids, tainted with uranyl acetate and lead citrate, and photographed using a JEOL JEM 1400 transmission electron microscope (JEOL, Peabody, MA) at 120 kV.

**Statistical analysis**. The data were analyzed using statistical analysis software (Statistical Analysis Software Institute). Grouped results from at least three different sets of experiments were expressed as mean with SEM, and differences between groups were assessed for significance by Student's $t$ test when the data were available in only two groups. When the data were available in more than two groups, analysis of variance (ANOVA) was used to perform pair-wise comparison. When significant main effects were detected ($P < 0.05$), Fisher least significant difference test was used to determine the differences between groups. A $P$ value of $< 0.05$ was considered to indicate statistical significance.

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

## Acknowledgements

We thank Lois Herrera for assistance in selected experiments. This work was supported by grants from the National Institutes of Health (HL68111 and ES015482 to Y.T.).

## Author contributions

Y.T. conceived the idea; Y.T. and Y.A.M. designed the research; Y.A.M., I.L.-B., N.L., M.G.W., H.S.C. performed research and analyzed the data; A.M.K.C. facilitated the electron microscopy studies; Y.T. and Y.A.M. wrote the manuscript with input from all authors.

## Additional information

**Competing interests:** The authors declare no competing financial interests.

