## [Peer Review File · Nature Communications]

Reviewers' Comments:

Reviewer #1 (Remarks to the Author)

MOMP and release of cytochrome c from mitochondria to initiate caspase activation requires oligomerization of Bax or Bak at the mitochondrial outer membrane. In intact cells, as opposed to synthetic membranes, additional events downstream of these oligomers are needed to support MOMP. This includes induction of a mitochondrial-ER platform that modulates Ca²⁺ transmission from ER stores into mitochondria, causing among other things cristae remodeling and mobilization of cytochrome c within the mitochondrial inter membrane space. Depending on the stimulus, Ca²⁺ overload in mitochondria can itself drive cell death. There may be multiple pathways that mediate these ER-mitochondrial events. One includes Bik, a BH3-only protein that localizes to ER, and has previously been shown to initiate such Ca²⁺ - mediated cristae remodeling, dependent on ER Bak. Here, Mebratu and co-authors extend this work and present evidence that Bik, via Bak, forms a complex with DAPk1 and ERK1/2 to increase contact of ER and mitochondria and facilitate Ca²⁺ uptake by mitochondria. Additionally, evidence is provided that a stapled helix corresponding to the Bik BH3 helix can initiate these Bak-dependent events at the ER, and in both cellular and animal models can resolve stimulus induced hyperplasia of airway epithelial cells. Overall, this is a large, very well conducted study employing multiple genetic approaches to uncover multiple new steps of the Bik pathway at the ER. There are however a number of experimental deficiencies and interpretation of results that will need to be addressed.

1. Throughout Fig. 1, the death status (caspase activation) of the cells was not determined and how this might influence the results. For example, do these upstream events occur in the presence of caspase inhibitors.
2. The language of the text leaves the impression that Bik is activating Bak even though there is no evidence that this BH3-only protein interacts with Bak. In contrast, the evidence presented indicates that Bik is inducing stabilization of Bak upon its release from Bcl-2. Since Bcl-2 de facto interacts with activated Bak (exposed BH3), the cells used in this study must be "primed". A critical experiment to support the model, therefore, is to demonstrate that over-expressed ER Bcl-2 (eg, Bcl-2b5) inhibits Bik initiated formation of the (B)DEB complex.
3. Fig. 2: protein levels were not presented in airway epithelial cells from Bak^{-/+} vs Bak^{+/+} animals. Also, what were the effects of Bak deletion (Bak^{-/-})?
4. The BDEB complex. A priori, there is no reason why Bik needs to be a part of the complex, especially if its primary role is to bind and displace other Bcl-2 binding partners at the ER. Moreover, the evidence presented suggests that interactions with DAPk1 are via the Bik BH3 helix, implying that DAPk1 has a Bcl-2-like binding groove. The only evidence for these interactions is via co-ip's of over-expressed proteins, which are strongly subject to artifacts. The authors need to validate these interactions using other means, including FP affinity measurements using synthetic BH3 peptides and purified domains of DAPk1, and compare to Bcl-2/Bik BH3 interactions.

Reviewer #2 (Remarks to the Author)

The manuscript by Mebratu et al describes how the BH3 domain of Bik upregulates the expression of Bak, then subsequently activates Bak and enriches it at the ER membrane. They also identify DAPk1 as an additional intracellular binding partner of Bik, which upon this interaction, forms a proposed ternary complex which also includes ERK and Bak. They then attempt to characterize the calcium mediated organelle signaling pathway that links compromised ER integrity with the mitochondria to induce cytochrome c release and subsequent caspase induced cell death. Overall this manuscript clearly exemplifies a substantial amount of work and effort by authors,

which is quite commendable in scope and breadth. The downside of this level of breadth is that the reader may feel overwhelmed with the amount of data that they are required to digest, as there are at least 90 figure subpanels in the main and supplemental texts. Within the page limits, this amount of data requires that the authors only are able to offer a terse description of each result. In one example, the authors describe the results of 7 panels of data in a single three line paragraph. The breadth of this study also results in this manuscript suffering from a degree of identity crisis. What is the main theme? Is it that Bik has distinct roles and signaling pathway functions in different cells types or that Bik expression reduces respiratory tract epithelium hyperplasia/metaplasia? Is the key finding that Bik directly interacts with Bak as opposed to displacing Bak from BCL2, or that Bik stabilizes Bak and prevents proteosomal degradation, or that DAPK1 is the long sought linkage between organelle ER/mito communication, or is that Bik interacts directly with and mediates IP3 related calcium efflux, or is that Bik stapled peptides might be translatable tools for studying and treating the resolution of hyperplastic mucous cells? Although the authors do allude to all these themes in the discussion—each of which could conceivably be a stand-alone manuscript—this reviewer believes, as currently written, the results are not sufficient to prove the authors many conclusions.

These gaps and deficiencies, some minor, some major, lead the reader to question the conclusions.

- The authors should be consistent with IFN or INF. Seems like IFN is in text but INF is in figures.
- For all western blot and IP, please note the relevant affinity tagged protein, for instance FLAG-IP should be FLAG-Dapk-IP.
- Please include full blots as supplemental materials.
- The authors cite Stout 2007 in arguing that Bax has a minimal role, yet in their 2007 paper, the authors state that IFN gamma causes translocation of Bax from cytosol to ER but not to mito. Similarly Tesfaigzi published a paper in 2002 stating the importance of Bax for IFN gamma induced resolution of allergen-induced mucus cell metaplasia. Granted, these are dated references and conclusions evolve, but the authors should resolve these discrepancies.
- Figure 1B: the authors claim this shows crucial importance of L61 for Bik induced Bak expression. The blot actually shows a tiny difference in expression, thus the role of L61 is uncertain (also need to include the IFN (-) lanes for each vector). Conversely in Supp1C, the authors claim that deleting NOXA has no role IFN induced increase in Bak expression. Again this difference borders on a slight increase. In both cases, the authors interpret the null to slight differences in IFN induced increase in Bak in the most favorable light that fits their conclusions. These blots need to be done in greater replicate and the band intensities quantified.
- The authors state " Bik expression did not affect Bak mRNA levels"... where is the data to support this?
- Figure 1D: shBCL2 seems to have no effect on BCL2 protein levels. Repeat expt with effective silencing of BCL2
- Figure 1E: are cells treated with IFN? shBCL2 is more effective in this blot. Combine D & E into 8 lane gel testing +/- effects of IFN & MG132 on shBCL2/shCtrl
- o Comparing the effects of silencing BCL2 and inhibiting proteasome, it is clear that proteosomal degradation dominates the effect, with minor contribution from BCL2.
- Figure 1F: how much Bik is produced by Ad-BIK? Is this comparable to native Bik in 1D, 3rd lane? If more than conclusion that only Bak bound by BCL2 is degraded is unproven.
- Supp1G: is this ER prep acceptable based on dim calnexin band on overexposed blot? Redo as 1H to look for laddering.
- For confocal quantification, what does colocalization mean? Shouldn't it be green+red=yellow... if so S1H clearly has most yellow, while its % colocalization is among the smallest at only 30%. Please explain. What is F3YpetR-F3hBak ?
- Figure 1K: does shBCL2 or MG132 induce Ca flux by confocal marker Fluor-4? Does expression of Bak Y108A make green color?
- Figure 1L and 1F: authors prove that overexpressed Bik is capable of disrupting Bak/BCL2 and IPR3/BCL2 complexes, yet BCL2 IP in 1L shows that Bik remains bound to BCL2. Seems that most of the results are geared towards proving that Bik interacts with Bak, yet this result seems to show that Bik interacting with BCL2, results in displacement of Bak.

- Figure 1M & S1M: this is the first data on mitochondrial Ca release. Does IFN induce Ca(m) release? How can the authors be sure that Bak activation by Bik results solely in effects at the ER and not at the mito?
- Figure 3A: this panel is suspect. The authors overexpress HA tagged Bik, yet the HA band in ERK IP is barely visible. Why switch to HA here as opposed to using Bik antibody in Fig1. Why not IP HA and look for all proposed interactors from this study, like Bak, BCL2, IP3R, DAPK1, Erk, etc? 3B seems more convincing, though, except that Bak has negligible decrease in interaction with Dapk upon overexpression of mutant Bik.
- Figure 3B, 3C, 3F: data supports conclusion that DAPK interacts most relevantly or directly with Bak. Figure 3I & J: DAPK activates Bak which causes caspase cleavage. Results do not support conclusion that DAPK interacts with Bak.
- Figure 4B bar graph says bak +/+ or -/-. Text doesn't mention bak. Is this typo?
- Supp 4A bar graph says fluorimeter was used to quantify intensity of fluor-4 and ER calcium. Which color is represented by bar graph? One or both, summed or averaged? More details needed.
- More explanation of the conclusion regarding the EM is needed. The authors show that IFN and Bik induce Bak mediated calcium and cyto c release by permeablizing the organelles. Are the authors claiming that the organelle disruption is what is leading to increased proximity? If not, how can this be ruled out? Also the authors state that DAPK1 is a large protein... are they claiming that this is sufficient to bridge the 20 nanometers between the organelles? Did the authors do immuno-gold staining of DAPK in their EM studies to substantiate this hypothesis?
- Were the stapled peptides tested for non-specific membrane disruption, ie LDH release?
- Please show LC and MS characterization of peptides.
- Cytotox of SHS1 not much better than scramble, while SHS2 is comparable to DHS, so likely conclusion is that cytotox of DHS driven by SHS2 while SHS1 contributes little. How were staple locations chosen?
- Sup5B: no real dose response. That is unexpected. Do peptides induce cell cycle arrest? Authors should test for this as well as apoptosis or caspase activation.
- Sup5C: strange now that SHS1 is more active than SHS2 in primary cells. Please comment on this.
- Figure 6B: vehicle treatment control for baseline Bak missing, likewise for Sup5E.
- Figure 6F: 50 milliliters (!!) of PBS
- Does stapled peptide disrupt Bak/Bik/Erk/DAPk ternary complex? Disrupt IP3R/BCL2 complex? Does stapled peptide induce Bak ER colocalization? Does stapled peptide treatment make green color by confocal Fluor-4 staining? Does biotin labeled stapled Bik peptide pulldown Bak?

Reviewer #3 (Remarks to the Author)

This manuscript by Mebratu examines the mechanisms by which the pro-apoptotic protein Bcl-2 interacting killer, Bik, elicits calcium release from ER stores. It follows the long-standing interest of the senior author in mucoid-cell hyperplasia in airways diseases. The manuscript demonstrates three mechanisms: 1) Bik dissociates Bak/Bcl-2 to rich for ER0-associated with Bak and interacts with DAPk1 to form a complex with Bak, 2) Bik disrupts an interaction between Bcl-2 and IP3R to elicit calcium release, and 3) ER associated Bak interacts with DAPk1 to increase contact between the R and mitochondria so as to facilitate mitochondrial calcium uptake. A Bik BH3 helix peptide was sufficient to elicit calcium release, and in mouse models of airway inflammation, reduced mucous cell hyperplasia. The authors suggest that Bik peptides may have therapeutic potential based on these mechanisms.

The multiplicity of techniques to demonstrate each major point is important and is convincing. The experiments to demonstrate that cells with lower DAPk1 levels were less sensitive to Bik-induced cell death with decreased Annexin-V positivity help to explain the previous findings of Coultas (2004).

Overall the methods are clear. As I note below a couple of the blots are not convincing, but the use of several techniques to illustrate the key points overcomes this.

Concerns:

1. The figure 1B showing the effect of Ad-Bik and Ad-Bik-L61G is not convincing. The difference in Bak expression is modestly less. Likewise, the increase in Bak expression in the face of IFN-g and shBcl-2 (Figure 1D) also is not convincing in a single blot. If these experiments have been repeated, showing densitometry measurements would be helpful.

Some explanation of the experiment presented in Figure 1H, with reference as to why the demonstration of oligomers is important, would help the reader.

2. Did allergen treatment in Bak +/- mice lead to evidence of epithelial damage other than epithelial cell hyperplasia? Allergen exposure frequently leads to loss of epithelial cells (perhaps due to apoptosis of other epithelial cell sub-types such as ciliated cells), gaps in the the basement membrane, etc.; did you see more of this in the Bak +/- mice?

3. Goblet cell hyperplasia is important in certain diseases of the colon: a demonstration of the extensibility of the work to include goblet cells other than airway would be useful. This need not be done in depth, but one or two key experiments to show that Bik regulates colon goblet cell apoptosis by similar mechanisms would have significant, additional impact for the work.

Minor concerns:

Figure 1D: Is it MG-132 and not MG-123?

Line 201: "...caspase 3 was by IFN-g..." It appears that 'activated' is missing.

Reviewer #1

MOMP and release of cytochrome c from mitochondria to initiate caspase activation requires oligomerization of Bax or Bak at the mitochondrial outer membrane. In intact cells, as opposed to synthetic membranes, additional events downstream of these oligomers are needed to support MOMP. This includes induction of a mitochondrial-ER platform that modulates Ca²⁺ transmission from ER stores into mitochondria, causing among other things cristae remodeling and mobilization of cytochrome c within the mitochondrial inter membrane space. Depending on the stimulus, Ca²⁺ overload in mitochondria can itself drive cell death. There may be multiple pathways that mediate these ER-mitochondrial events. One includes Bik, a BH3-only protein that localizes to ER, and has previously been shown to initiate such Ca²⁺ - mediated cristae remodeling, dependent on ER Bak. Here, Mebratu and co-authors extend this work and present evidence that Bik, via Bak, forms a complex with DAPk1 and ERK1/2 to increase contact of ER and mitochondria and facilitate Ca²⁺ uptake by mitochondria. Additionally, evidence is provided that a stapled helix corresponding to the Bik BH3 helix can initiate these Bak-dependent events at the ER, and in both cellular and animal models can resolve stimulus induced hyperplasia of airway epithelial cells. Overall, this is a large, very well conducted study employing multiple genetic approaches to uncover multiple new steps of the Bik pathway at the ER. There are however a number of experimental deficiencies and interpretation of results that will need to be addressed.

C1. Throughout Fig. 1, the death status (caspase activation) of the cells was not determined and how this might influence the results. For example, do these upstream events occur in the presence of caspase inhibitors.

R1. IFN- γ and Bik cause cell death in epithelial cells in a caspase dependent manner. We have shown in Fig. S3C and D that IFN- γ and Bik activate caspase 3 and this caspase activation occurs in a DAPk-dependent manner (Fig. 3I and J). Selected experiments to investigate ER-Ca²⁺ release, mitochondrial Ca²⁺ accumulation, and cytochrome c release assays were performed in the presence of the pan-caspase inhibitor, Q-VD-OPh, to avoid loss of cells. Inhibition of caspases did not affect the Bik-initiated ER-Ca²⁺ release and mitochondrial Ca²⁺ accumulation (Fig. 1J), Bak activation (S1H, S1I, S1K), or cytochrome c release (Fig. 3H). These data show that the upstream events are not affected by the presence of caspase inhibitors as caspases are downstream of cytochrome c release from mitochondria.

C2. The language of the text leaves the impression that Bik is activating Bak even though there is no evidence that this BH3-only protein interacts with Bak. In contrast, the evidence presented indicates that Bik is inducing stabilization of Bak upon its release from Bcl-2. Since Bcl-2 de facto interacts with activated Bak (exposed BH3), the cells used in this study must be “primed”. A critical experiment to support the model, therefore, is to demonstrate that over-expressed ER Bcl-2 (eg, Bcl-2b5) inhibits Bik initiated formation of the (B)DEB complex.

R2. We have now over-expressed ER-targeted Bcl-2 (Bcl-2b5) in HEK293T cells and to investigate whether Bik-induced cell death and the formation of BDEB complex is affected. We find that expression of Bcl-2b5 reduced Bak levels and Bik-induced cell death (Fig. 3L) and impaired BDEB formation (Fig. 3M). These results support the model that Bik dissociates Bcl-2 from Bik, so that Bak oligomerization occurs at the ER to bind to DAPk1 and facilitate ER-Ca²⁺ transfer to the mitochondria.

C3. Fig. 2: protein levels were not presented in airway epithelial cells from Bak^{-/+} vs Bak^{+/+} animals. Also, what were the effects of Bak deletion (Bak^{-/-})?

R3: We have now included a Western blot showing the levels of Bak in *bak^{+/-}* and *bak^{+/+}* mouse airway epithelial cells (Fig. 2E). Fig. 2A and 2B show that deletion of Bak protects mouse airway epithelial cells from IFN- γ - or Bik-induced cell death, and in Fig. 2C and D, we have shown that reconstituting Bak in *bak^{-/-}* cells sensitizes cells to Bik-induced cell death. Collectively, these *in vitro* and *in vivo* data show that the amount of activated ER Bak is a determinant for the extent of ER and mitochondria proximity across the cell. The resulting frequency of contact sites between ER and mitochondria determine extent of MOMP disruption and cytochrome c release that ultimately activates downstream caspases. (see Discussion, page 17, line 6).

C4. The evidence presented suggests that interactions with DAPK1 are via the Bik BH3 helix, implying that DAPK1 has a Bcl-2-like binding groove. The only evidence for these interactions is via co-ip's of over-expressed proteins, which are strongly subject to artifacts. The authors need to validate these interactions using other means, including FP affinity measurements using synthetic BH3 peptides and purified domains of DAPK1, and compare to Bcl-2/Bik BH3 interactions.

R4: As mentioned by this and other Reviewers, the presented studies suggest that there is a direct interaction of Bik with DAPK1. Whether the nature of this interaction involves a Bcl-2-like binding groove is beyond the scope of this manuscript. However, we will pursue these studies in the near future to confirm and understand the molecular interaction of Bik with the kinase domain of DAPK1.

Reviewer #2

C1: Overall this manuscript clearly exemplifies a substantial amount of work and effort by authors, which is quite commendable in scope and breadth. The downside of this level of breadth is that the reader may feel overwhelmed with the amount of data that they are required to digest, as there are at least 90 figure subpanels in the main and supplemental texts. Within the page limits, this amount of data requires that the authors only are able to offer a terse description of each result. In one example, the authors describe the results of 7 panels of data in a single three line paragraph. The breadth of this study also results in this manuscript suffering from a degree of identity crisis. What is the main theme? Is it that Bik has distinct roles and signaling pathway functions in different cells types or that Bik expression reduces respiratory tract epithelium hyperplasia/metaplasia? Is the key finding that Bik directly interacts with Bak as opposed to displacing Bak from BCL2, or that Bik stabilizes Bak and prevents proteosomal degradation, or that DAPK1 is the long sought linkage between organelle ER/mito communication, or is that Bik interacts directly with and mediates IP3 related calcium efflux, or is that Bik stapled peptides might be translatable tools for studying and treating the resolution of hyperplastic mucous cells?

R1: As Reviewer 1 stated the many experiments presented in this manuscript show that “Bik, via Bak, forms a complex with DAPK1 and ERK1/2 to increase contact of ER and mitochondria and facilitate Ca²⁺ uptake by mitochondria. Additionally, evidence is provided that a stapled helix corresponding to the Bik BH3 helix can initiate these Bak-dependent events at the ER, and in both cellular and animal models can resolve stimulus induced hyperplasia of airway epithelial cells.” The molecular findings are shown to be valid *in vivo* in the context of resolution of

epithelial cell hyperplasia. Multiple genetic approaches had to be employed to uncover multiple new steps of the Bik pathway at the ER. Therefore, we do not believe that the manuscript lacks a main theme. The main theme is to uncover the role(s) of Bik in ultimately causing death in epithelial cells that has a major role in maintaining a constant number of cells in airways.

C2: Although the authors do allude to all these themes in the discussion—each of which could conceivably be a stand-alone manuscript—this reviewer believes, as currently written, the results are not sufficient to prove the authors many conclusions. These gaps and deficiencies, some minor, some major, lead the reader to question the conclusions.

R2: We do not believe that the findings addressing the various functions of Bik in causing cell death should be presented individually. Bik fulfills 3 distinct roles – disrupt Bcl-2-Bak interaction to stabilize Bak, prime DAPk1 to form the BDEB complex and tether ER and mitochondria, and dissociate Bcl-2 from IP₃R to facilitate ER-Ca²⁺ release, all leading to disrupt MOMP.

C3: The authors should be consistent with IFN or INF. Seems like IFN is in text but INF is in figures.

R3. We have now changed INF to IFN in the text and figures. See figures 1A, C, D, H, 2A, 3I, 4C. In supplement, figures S1A, B, E, F, G, H, I, K, and N, and S2A.

C4: For all western blot and IP, please note the relevant affinity tagged protein, for instance FLAG-IP should be FLAG-Dapk-IP.

R4: We have now included the relevant affinity tagged protein for all Western and IP blots in which affinity tagged proteins have been used. See figures 3B, 3C, 3D, 3M.

C5: Please include full blots as supplemental materials.

R5: We have included full blots in a separate file as Supplemental Materials.

C6: The authors cite Stout 2007 in arguing that Bax has a minimal role, yet in their 2007 paper, the authors state that IFN gamma causes translocation of Bax from cytosol to ER but not to mito. Similarly Tesfaigzi published a paper in 2002 stating the importance of Bax for IFN gamma induced resolution of allergen-induced mucus cell metaplasia. Granted, these are dated references and conclusions evolve, but the authors should resolve these discrepancies.

R6: Our earlier studies found that IFN- γ -induced airway epithelial cell death involves the ER. We found that *bax*^{+/+} and *bax*^{-/-} mouse airway epithelial cells showed minor differences in IFN- γ -induced cell death.¹ However, when investigating the role of Bak using *bak*^{+/+} and *bak*^{-/-} MAECs the protection by reduced Bak levels were far greater than by deficiency in Bax. This point is now included in the Discussion (page 15, line 1).

C7: Figure 1B: the authors claim this shows crucial importance of L61 for Bik induced Bak expression. The blot actually shows a tiny difference in expression, thus the role of L61 is uncertain (also need to include the IFN (-) lanes for each vector). Conversely in Supp1C, the authors claim that deleting NOXA has no role IFN induced increase in Bak expression. Again this difference borders on a slight increase. In both cases, the authors interpret the null to slight differences in IFN γ -induced increase in Bak in the most favorable light that fits their conclusions. These blots need to be done in greater replicate and the band intensities quantified.

R7: As suggested, we have repeated these experiments to confirm that Bak levels are increased robustly and consistently. Furthermore, we have included densitometry quantification to show the fold change in Bak expression (Fig 1B).

C8: The authors state “ Bik expression did not affect Bak mRNA levels”... where is the data to support this?

R8: We have now included the data on Bak mRNA expression data in Fig. S1G, showing that Bak mRNA levels are increased in both *bik*^{+/+} and *bik*^{-/-} MTECs by IFN- γ treatment.

C9: Figure 1D: shBCL2 seems to have no effect on BCL2 protein levels. Repeat expt with effective silencing of BCL2. Figure 1E: are cells treated with IFN? shBCL2 is more effective in this blot. Combine D & E into 8 lane gel testing +/- effects of IFN & MG132 on shBCL2/shCtrl. Comparing the effects of silencing BCL2 and inhibiting proteasome, it is clear that proteosomal degradation dominates the effect, with minor contribution from BCL2.

R9: We have repeated the experiment and combined figures 1D and E to show that silencing Bcl-2 leads to increased level of Bak. We also have included densitometry quantification of Bak levels to show changes in the expression of Bak protein. In this experiment, cells treated with MG-132 were not treated with IFN- γ , but also showed increased Bak levels. In addition to silencing of Bcl-2, as also noted by the Reviewer, inhibiting proteasomal degradation increases Bak levels, but suppression of Bcl-2 expression further enhances stability of Bak. These findings suggest that Bcl-2 may promote Bak degradation by mechanisms that do not involve the proteasome (see Discussion on page15, line 23).

C10: Figure 1F: how much Bik is produced by Ad-BIK? Is this comparable to native Bik in 1D, 3rd lane? If more than conclusion that only Bak bound by BCL2 is degraded is unproven.

R10: Fig. 1E (1F in the former version) now shows the level of Bik in the input of Ad-Bik or Ad Bik^{L61G} expressing cells. The Bik levels observed in cells infected with 100 MOI Ad-Bik (Fig. 1F) are comparable to those when cells are treated with 50 ng/ml IFN- γ shown in Fig. 1D (1E in the former version).

C11: Supp1G: is this ER prep acceptable based on dim calnexin band on overexposed blot? Redo as 1H to look for laddering.

R11: We repeated the experiment and have replaced the Western blot in Fig.S1J (S1H in the former version). The new figure shows a clear band for calnexin.

C12: For confocal quantification, what does colocalization mean? Shouldn't it be green+red=yellow... if so S1H clearly has most yellow, while its % colocalization is among the smallest at only 30%. Please explain.

R12: The yellow color indicates co-localization of the red and green colors, representing Bak and calnexin detection, respectively. The percent co-localization was quantified by counting 200 cells for each experiment and averaging the data from at least 3 independent experiments. The photomicrograph in S1K (S1H in previous version) shows only a representative cell displaying the co-localization of Bak and calnexin.

C13: What is F3YpetR-F3hBak ?

R13: F3YpetR-F3hBak is fluorescent protein-tagged Bak construct we used to track expressed Bak as it translocates to the ER in response to Bik expression. The fluorescence of F3YpetR-F3h is shown in green and calnexin is shown in red. The areas where ER-Bak and calnexin co-localize are shown in yellow that was quantified by Manders' coefficient. This fluorescent Bak construct was also tagged with triple Flag-tagged cYpet, as described in the Supplement (page 3 bottom).

C14: Figure 1K: does shBCL2 or MG132 induce Ca flux by confocal marker Fluor-4? Does expression of Bak Y108A make green color?

R14: Fig. 1J shows that IFN- γ and Bik lead to ER Ca²⁺ release, and Fig. 1L shows that Bak mediates this Ca²⁺ release. Because, the focus of this manuscript is to elucidate the role of Bik in inducing cell death, studies on the effect of shBcl-2 or MG-132 would not directly clarify the role of Bik, rather would distract from the main focus of the paper.

C15: Figure 1L and 1F: authors prove that overexpressed Bik is capable of disrupting Bak/BCL2 and IPR3/BCL2 complexes, yet BCL2 IP in 1L shows that Bik remains bound to BCL2. Seems that most of the results are geared towards proving that Bik interacts with Bak, yet this result seems to show that Bik interacting with BCL2, results in displacement of Bak.

R15: As the Reviewer clarified, the results show that Bik binds to Bcl-2 to displace IP₃R-1 from Bcl-2. The findings suggest that Bik has additional functions that are independent of IP₃R-1. It also displaces Bak from Bcl-2, and leads to stabilization, activation, and ultimately translocation to the ER. The focus of this paper is to show that Bik causes cell death by affecting three independent functions. This is now summarized in a schematic (Fig. 7).

C16: Figure 1M & S1M: this is the first data on mitochondrial Ca release. Does IFN induce Ca(m) release? How can the authors be sure that Bak activation by Bik results solely in effects at the ER and not at the mito?

R16: Figure S1P (S1M in previous version) shows that Bik causes mitochondrial calcium accumulation in a Bak-dependent manner. Based on previous studies^{2,3} we have to assume that ER-Ca²⁺ precedes the accumulation of mitochondrial Ca²⁺. Our studies also support the idea that ER-Bak is the main driver of ER-Ca²⁺ release that results in mitochondrial Ca²⁺ accumulation. (see Discussion page 16, line 16)

C17: Figure 3A: this panel is suspect. The authors overexpress HA tagged Bik, yet the HA band in ERK IP is barely visible. Why switch to HA here as opposed to using Bik antibody in Fig1. Why not IP HA and look for all proposed interactors from this study, like Bak, BCL2, IP3R, DAPK1, Erk, etc? 3B seems more convincing, though, except that Bak has negligible decrease in interaction with Dapk upon overexpression of mutant Bik.

R17: In experiment 3A (also all other experiments in Fig.3) we used HA tagged Ad-Bik to investigate whether the overexpressed exogenous Bik interacts with DAPK1 and ERK. We have now improved the quality of the Western in Fig. 3A with enhanced visibility of the HA-tagged Bik signal. We did HA-Bik IP in cells expressing different domains of DAPK1 in Fig. 3C to avoid potential signal from endogenous Bik. These studies show that the HA-Bik interacts with the kinase domain of DAPK1 (GFP-DAPK1).

C18: Figure 3B, 3C, 3F: data supports conclusion that DAPK interacts most relevantly or directly with Bak. Figure 3I & J: DAPK activates Bak which causes caspase cleavage. Results do not support conclusion that DAPK interacts Bik.

R18: Fig. 3B shows HA-tagged Bik interacts with Flag-tagged DAPK. Similarly, Fig. 3C shows that the kinase domain of DAPK1 interacts with HA-tagged Bik. Further, Fig 3D shows that while the WT DAPK1 binds to HA-Bik, K42A mutant did not. Therefore, these results clearly support that DAPK1 interacts with Bik, and mutation of the kinase domain impaired this interaction. See also Discussion, page 16, 2nd paragraph.

C19: Figure 4B bar graph says bak +/- or -/-. Text doesn't mention bak. Is this typo?

R19: The labeling on the y-axis of figure 4B was a typo and has now been corrected.

C20: Supp 4A bar graph says fluorimeter was used to quantify intensity of fluo-4 and ER calcium. Which color is represented by bar graph? One or both, summed or averaged? More details needed.

R20: Fluo-4 is a marker for ER-Ca²⁺ release. The fluorimeter is used to measure the intensity of Fluo-4 to indicate the level of ER Ca²⁺ release. Therefore, the bar graph represents the green color, which denotes release of ER-Ca²⁺. As stated in the figure legend, the quantification is the average of at least 3 independent experiments.

C21: More explanation of the conclusion regarding the EM is needed. The authors show that IFN and Bik induce Bak mediated calcium and cyto c release by permeablizing the organelles. Are the authors claiming that the organelle disruption is what is leading to increased proximity? If not, how can this be ruled out? Also the authors state that DAPK1 is a large protein... are they claiming that this is sufficient to bridge the 20 nanometers between the organelles? Did the authors do immuno-gold staining of DAPK in their EM studies to substantiate this hypothesis?

R21: Our finding suggests that the assembly of the BDEB complex leads to the proximity of ER and mitochondria. Merely, suppression of DAPK 1 levels (Fig. 4E) or reduced Bak levels as found in *bak*^{+/-} MAECs (Fig. 2F, G) abrogate ER-Ca²⁺ release and cell death, respectively. Therefore, we conclude that the tethering of mitochondria and ER precede organelle disruption. To add immune-gold staining to the data presented is beyond the scope of this manuscript.

C22: Were the stapled peptides tested for non-specific membrane disruption, ie LDH release?

R22: HC-stapled peptides of the bcl-2 family protein have been used in a number of studies from other laboratories^{4,5} with no report of membrane disruption. Additionally, the mutant Bik peptide serves as controls for all other experiments. We did not observe cell death (trypan blue exclusion assay) with mutant peptides.

C23: Please show LC and MS characterization of peptides.

R23: The LC and MS characterization of the peptides are provided in an additional Supplemental file.

C24: Cytotox of SHS1 not much better than scramble, while SHS2 is comparable to DHS, so likely conclusion is that cytotox of DHS driven by SHS2 while SHS1 contributes little. How were staple locations chosen?

R24: Because all of our studies showed that the BH-3 domain is the active site, we designed several peptides that comprise the BH-3 domain. Previous studies have demonstrated that modification of these peptides by formation of hydrocarbon-staples, referred to as stabilized alpha-helix of Bcl-2 domains (SAHBs), stabilizes helical conformation and increases stability⁵

for Bim⁶, Bid⁴, and Bad⁷ BH3 domains. We designed the substitution of amino acids to (S)-2-(2'-pentenyl) Ala residues based on 3-D structures for Bad.⁷ Peptides were also labeled with the fluorescent tag carboxyfluorescein (FAM) to better monitor uptake into cells. Bik-derived peptides with single or double staples and the mutant controls (Leu changed to Gly) were synthesized to demonstrate that a single amino acid substitution is sufficient to discriminate in the killing activity. In addition, a peptide with scrambled amino acid sequence was prepared as further control. This is now added to Supplemental Data, page 4.

C25: Sup5B: no real dose response. That is unexpected. Do peptides induce cell cycle arrest? Authors should test for this as well as apoptosis or caspase activation.

R25: The data in S5B show a dose-dependent response from 0, 1, to 2.5 μ M that leveled off at the doses of 2.5, 5, and 10 μ M. The effect of these peptides on cell cycle arrest is not expected, as such function has not been reported for Bik in other studies.⁸ All our findings point to the fact that Bik peptides cause apoptotic cell death in the airway epithelial cell *in vivo* and in culture.

C26: Sup5C: strange now that SHS1 is more active than SHS2 in primary cells. Please comment on this.

R26: The difference in cell death between SHS1 and SHS2 was not statistically significant from the scramble peptide (Fig. S5C). In primary human airway epithelial cells DHS-Bik^{WT} was the only peptide which caused statistically significant cell death.

C27: Figure 6B: vehicle treatment control for baseline Bak missing, likewise for Sup5E.

R27: In addition to Bik^{L61G}, we have now included the vehicle control in the revised manuscript (Fig. 6B).

C28: Figure 6F: 50 milliliters (!!) of PBS

R28: The typo is corrected to 50 μ M.

C29: Does stapled peptide disrupt Bak/Bik/Erk/DAPk ternary complex? Disrupt IP3R/BCL2 complex? Does stapled peptide induce Bak ER colocalization? Does stapled peptide treatment make green color by confocal Fluor-4 staining? Does biotin labeled stapled Bik peptide pulldown Bak?

R29: We have shown in Fig. 6B that stapled Bik peptide increases the level of Bak protein. We further showed that the staple Bik peptide displaces Bak from Bcl-2 (Fig. 6C). Staple Bik peptides also caused Bak ER colocalization (Fig. 6G) and TUNEL positivity (Fig. 6H) in the airways of mice *in vivo* (Fig. 6G). These findings support the fact that stapled Bik peptides cause cell death in the airway epithelial cells by the same pathway as the Bik protein. More importantly, the fact that the Bik peptide while lacking the C-terminal ER anchoring domain acts like the whole Bik protein, suggests, that it is not Bik but ER-Bak that anchors DAPk1 to the ER (see Discussion page 17, line 18).

Reviewer #3

This manuscript by Mebratu examines the mechanisms by which the pro-apoptotic protein Bcl-2 interacting killer, Bik, elicits calcium release from ER stores. It follows the long-standing interest of the senior author in mucoid-cell hyperplasia in airways diseases. The manuscript

demonstrates three mechanisms: 1) Bik dissociates Bak/Bcl-2 to rich for ER-associated with Bak and interacts with DAPk1 to form a complex with Bak, 2) Bik disrupts an interaction between Bcl-2 and IP3R to elicit calcium release, and 3) ER associated Bak interacts with DAPk1 to increase contact between the ER and mitochondria so as to facilitate mitochondrial calcium uptake. A Bik BH3 helix peptide was sufficient to elicit calcium release, and in mouse models of airway inflammation, reduced mucous cell hyperplasia. The authors suggest that Bik peptides may have therapeutic potential based on these mechanisms. The multiplicity of techniques to demonstrate each major point is important and is convincing. The experiments to demonstrate that cells with lower DAPk1 levels were less sensitive to Bik-induced cell death with decreased Annexin-V positivity help to explain the previous findings of Coultas (2004).

Overall the methods are clear. As I note below a couple of the blots are not convincing, but the use of several techniques to illustrate the key points overcomes this.

Concerns:

C1. The figure 1B showing the effect of Ad-Bik and Ad-Bik-L61G is not convincing. The difference in Bak expression is modestly less. Likewise, the increase in Bak expression in the face of IFN-g and shBcl-2 (Figure 1D) also is not convincing in a single blot. If these experiments have been repeated, showing densitometry measurements would be helpful.

R1: We have now repeated and replaced the Western in Fig. 1B. Regarding figure 1D, as was also suggested by Reviewer 2, we repeated the experiment and have now combined figures 1D and E as Figure 1D. We have also included densitometry for both figures to demonstrate that increases in Bak protein level are consistent.

C2: Some explanation of the experiment presented in Figure 1H, with reference as to why the demonstration of oligomers is important, would help the reader.

R2: When Bak undergoes the N-terminal conformational change the BH3 domain is exposed⁹ and Bak multimerizes¹⁰. The present studies suggest that Bik-induced activation of Bak leads to Bak oligomerization at the ER. The oligomerized Bak anchors DAPk1 to the ER and provides the means for the released ER Ca²⁺ to enter mitochondria that are in close proximity. This is now included in Discussion (page 15, line 10 and 14).

C3: Did allergen treatment in Bak +/- mice lead to evidence of epithelial damage other than epithelial cell hyperplasia? Allergen exposure frequently leads to loss of epithelial cells (perhaps due to apoptosis of other epithelial cell sub-types such as ciliated cells), gaps in the basement membrane, etc.; did you see more of this in the Bak +/- mice?

R3: We have not observed any evidence of gaps in the basement membrane or damage to epithelium in *bak*^{+/+} or *bak*^{-/-} mice when the resolution of epithelial cell hyperplasia occurs over a period of time in *bak*^{+/+} mice. Detailed mechanisms of the removal process of dying cells from the epithelium will be studied in the future (Discussion page 18, line 5).

C4: Goblet cell hyperplasia is important in certain diseases of the colon: a demonstration of the extensibility of the work to include goblet cells other than airway would be useful. This need not be done in depth, but one or two key experiments to show that Bik regulates colon goblet cell apoptosis by similar mechanisms would have significant, additional impact for the work.

R4: This manuscript has used colon epithelial cell line (HCT116) that are either wild-type or deficient in Bax and Bak in figures 4F and S1M to investigate the molecular mechanism of Bik-induced cell death. Previous studies showed the Bik plays a major role in regulating colon epithelial cell death.¹¹ In addition, as suggested by this Reviewer, we now added a study to show that Bik causes increases in Bak expression (Fig. S1C) and cell death (Fig. S1D) in primary mouse colonic epithelial cells. Therefore, the findings suggests that this approach may be useful for treating diseases associated with epithelial cell hyperplasia other than in airways (see Discussion page 18, line 13).

Minor concerns:

Figure 1D: Is it MG-132 and not MG-123?

We have now replaced 'MG-123' with MG-132 in Fig. 1D.

Line 201: "...caspase 3 was by IFN-g..." It appears that 'activated' is missing.

We have now added the word "activated"

References

1. Stout BA, Melendez K, Seagrave J, Holtzman MJ, Wilson B, Xiang J, *et al.* STAT1 activation causes translocation of Bax to the endoplasmic reticulum during the resolution of airway mucous cell hyperplasia by IFN-gamma. *J Immunol* 2007, **178**(12): 8107-8116.
2. Orrenius S, Zhivotovsky B, Nicotera P. Regulation of cell death: the calcium-apoptosis link. *Nature reviews Molecular cell biology* 2003, **4**(7): 552-565.
3. Clapham DE. Calcium signaling. *Cell* 2007, **131**(6): 1047-1058.
4. Walensky LD, Pitter K, Morash J, Oh KJ, Barbuto S, Fisher J, *et al.* A stapled BID BH3 helix directly binds and activates BAX. *Molecular cell* 2006, **24**(2): 199-210.
5. Walensky LD, Kung AL, Escher I, Malia TJ, Barbuto S, Wright RD, *et al.* Activation of apoptosis in vivo by a hydrocarbon-stapled BH3 helix. *Science* 2004, **305**(5689): 1466-1470.
6. Labelle JL, Katz SG, Bird GH, Gavathiotis E, Stewart ML, Lawrence C, *et al.* A stapled BIM peptide overcomes apoptotic resistance in hematologic cancers. *The Journal of clinical investigation* 2012, **122**(6): 2018-2031.

7. Danial NN, Walensky LD, Zhang CY, Choi CS, Fisher JK, Molina AJ, *et al.* Dual role of proapoptotic BAD in insulin secretion and beta cell survival. *Nature medicine* 2008, **14**(2): 144-153.
8. Mebratu YA, Dickey BF, Evans C, Tesfaigzi Y. The BH3-only protein Bik/Blk/Nbk inhibits nuclear translocation of activated ERK1/2 to mediate IFN γ -induced cell death. *The Journal of cell biology* 2008, **183**(3): 429-439.
9. Dewson G, Kratina T, Sim HW, Puthalakath H, Adams JM, Colman PM, *et al.* To trigger apoptosis, Bak exposes its BH3 domain and homodimerizes via BH3:groove interactions. *Molecular cell* 2008, **30**(3): 369-380.
10. Wei MC, Lindsten T, Mootha VK, Weiler S, Gross A, Ashiya M, *et al.* tBID, a membrane-targeted death ligand, oligomerizes BAK to release cytochrome c. *Genes & development* 2000, **14**(16): 2060-2071.
11. Kovi RC, Paliwal S, Pande S, Grossman SR. An ARF/CtBP2 complex regulates BH3-only gene expression and p53-independent apoptosis. *Cell death and differentiation* 2010, **17**(3): 513-521.

Reviewers' Comments:

Reviewer #1:

Remarks to the Author:

The authors have satisfactorily addressed my concerns with one important exception: the question of Bik being part of the DAPk1-ERK1/2-Bak complex once Bik initiates release of activated Bak, which (presumably) oligomerizes and drives its interaction with Dapk1. In this model under physiological conditions, the levels of Bik would need to be high enough to BOTH associate with Bcl-2 to displace Bak and be in sufficient excess to also interact with the DAPk1-Bak complex. Their conclusion was based on a single co-ip with over-expressed Bik. The authors now indicate that my original proposal to verify the presence of Bik in the complex by another validating approach was beyond the scope of the present study. While I remain very supportive of this study overall, I would encourage the authors, at the minimum, to qualify the conclusion/language regarding Bik's presence in the Bak-DAPk1 complex.

Reviewer #2:

None

Reviewer #3:

Remarks to the Author:

No further comments to the author.

REVIEWERS' COMMENTS:

Reviewer #1 (Remarks to the Author):

The authors have satisfactorily addressed my concerns with one important exception: the question of Bik being part of the DAPk1-ERK1/2-Bak complex once Bik initiates release of activated Bak, which (presumably) oligomerizes and drives its interaction with Dapk1. In this model under physiological conditions, the levels of Bik would need to be high enough to BOTH associate with Bcl-2 to displace Bak and be in sufficient excess to also interact with the DAPk1-Bak complex. Their conclusion was based on a single co-ip with over-expressed Bik. The authors now indicate that my original proposal to verify the presence of Bik in the complex by another validating approach was beyond the scope of the present study. While I remain very supportive of this study overall, I would encourage the authors, at the minimum, to qualify the conclusion/language regarding Bik's presence in the Bak-DAPk1 complex.

Response by Authors:

We agree that the interaction with DAPk1 via the Bik BH3 helix is based on the co-ip of overexpressed DAPk1 and Bik. Therefore, we have now added to the Discussion (page 16, line 13): At present, the only evidence for Bik and DAPk1 interaction is based on co-immunoprecipitation of over-expressed Bik and future studies will validate this finding using other approaches such as fluorescent polarization assay. These studies would confirm the possibility that DAPk1 may have a Bcl-2-like binding groove in the kinase domain.